# Distribution and sources of fallout $^{137}$Cs and $^{239+240}$Pu in Equatorial and Southern Hemisphere reference soils

Gerald Dicen[1,2], Floriane Guillevic[1], Surya Gupta[1], Pierre-Alexis Chaboche[3,4,5], Katrin Meusburger[6], Pierre Sabatier[7], Olivier Evrard[5], Christine Alewell[1]

[1]Environmental Geosciences, Department of Environmental Science, University of Basel, Bernoullistrasse 30, 4056 Basel, Switzerland

[2]Department of Science and Technology-Philippine Nuclear Research Institute (DOST-PNRI), Commonwealth Avenue, Diliman, 1101 Quezon City, Philippines

[3]International Research Fellow of Japan Society for the Promotion of Science (Postdoctoral Fellowships for Research in Japan (Standard)), Japan

[4]Institute of Environmental Radioactivity, Fukushima University, Kanayagawa, Japan

[5]Laboratoire des Sciences du Climat et de l'Environnement (LSCE/IPSL), Unité Mixte de Recherche 8212 (CEA-CNRS-UVSQ), Université Paris-Saclay, F-91191 Gif-sur-Yvette, France

[6]Swiss Federal Institute for Forest Snow and Landscape Research WSL, Birmensdorf, Zürich, Switzerland

[7]Université Savoie Mont-Blanc, CNRS, EDYTEM, F-73000 Chambéry, France

*Correspondence to:* Gerald Dicen (gerald.dicen@unibas.ch)

**Abstract.** Past nuclear weapons testing (NWT) and nuclear power plant (NPP) accidents have resulted in the ubiquitous deposition of radionuclides in the environment. These fallout radionuclides (FRNs) are considered the privileged markers ("golden spikes") of the Anthropocene stratigraphic layers. Their deposition in the 1950s coincided with the "Great Acceleration", which is characterized by large-scale shifts in the Earth's systems, including increased land-use change and soil degradation. Among the FRNs deposited globally, $^{137}$Cs has been the most commonly used to assess soil erosion and/or the chronology of sediment deposition, and $^{239+240}$Pu is an alternative soil erosion tracer and chronological marker increasingly used due to a number of advantages.

We compiled $^{137}$Cs and $^{239+240}$Pu data published from undisturbed (so called "reference") soils in the Equatorial and Southern Hemisphere regions to build a database under the AVATAR Project ("A reVised dATing framework for quantifying geomorphological processes during the Anthropocene"). Using this database, named the AVATAR-Soils Database, we determined the distribution of $^{137}$Cs and $^{239+240}$Pu inventories in Equatorial and Southern Hemisphere soils, along with the relative contributions of different fallout nuclear weapon sources by analysing their isotopic ratios. Additionally, we demonstrated how the database can be used to identify the environmental factors that influence the distribution of $^{137}$Cs and $^{239+240}$Pu in reference soils by applying a machine-learning algorithm.

Our metanalysis revealed that high $^{137}$Cs and $^{239+240}$Pu inventories were recorded near the equator and within the 20-40° S latitudinal bands, which coincide with the location of multiple NWT. The $^{240}$Pu/$^{239}$Pu atom ratios suggest that sources other than the global fallout (primarily from US and USSR weapon testing with a $^{240}$Pu/$^{239}$Pu atom ratio of ~ 0.18) contributed to the reference inventories in the Southern Hemisphere. These additional sources have been relatively neglected so far. Based on the $^{240}$Pu/$^{239}$Pu atom ratios, we have estimated that the French fallout contributed ~20% to the reference soil $^{239+240}$Pu inventories in South America and up to 70% in French Polynesia. In contrast, the British fallout contributed ~27% to the reference soil $^{239+240}$Pu inventories in the rest of Oceania. Our machine-learning algorithm identified precipitation of the coldest quarter, longitude, and latitude as the strongest predictors of $^{137}$Cs inventory. For $^{239+240}$Pu inventory, mean diurnal temeperature range, temperature annual range, and precipiation of the driest quarter were the strongest predictors. Altogether, these findings demonstrate the potential of the AVATAR-Soils Database as resource for improving our understanding of the distribution and sources of $^{137}$Cs and $^{239+240}$Pu in Equatorial and Southern Hemisphere soils and refining their application as tools in various Earth Science research. The AVATAR-Soils Database may be accessed at https://doi.org/10.5281/zenodo.14008220 (Dicen et al., 2024).

# 1. Introduction

## 1.1 Background

Radionuclide deposition from nuclear weapons testing (NWT) and nuclear power plant (NPP) accidents has become a global concern. Exposure to radionuclides can potentially lead to health problems and adverse ecological impacts because of the radiation hazards they pose and the toxicity of many of these radionuclides (Owens et al., 2019; Bouville, 2020). Radionuclides are therefore considered as serious environmental contaminants, especially those with long residence times. Despite the potential risks associated with them, fallout radionuclides (FRNs) with long half-lives provide the privileged markers ("golden spikes") of the Anthropocene stratigraphic layers (Certini and Scalenghe, 2021). The onset of their deposition in the 1950s coincided with the "Great Acceleration", which is characterized by large-scale shifts in the biophysical and socioeconomic components of the Earth System (Steffen et al., 2015), including an increase in soil degradation, mainly triggered by land-use change (Ferraro et al., 2018; Wang et al., 2022). The concentrations of FRNs and their isotopic ratios were shown to provide reliable indicators of contamination sources (Alewell et al., 2014; Meusburger et al., 2020; Evrard et al., 2023), environmental impacts (Steinhauser et al., 2014; Foucher et al., 2023), sediment chronology (Bruel & Sabatier, 2020), soil dating (Ferreira et al., 2016), soil redistribution (Alewell et al., 2017; Mabit et al., 2008), and particle transfers within soil profiles (Jagercikova et al., 2015), and therefore represent a major interest in Earth Science research. Commonly studied FRNs include $^{241}$Am ($t_{1/2}$ = 432.6 years), $^{129}$I ($t_{1/2}$ = $1.614 \times 10^7$ years), $^{14}$C ($t_{1/2}$ = 5,730 years) , $^3$H ($t_{1/2}$ = 12.32 years), $^{137}$Cs ($t_{1/2}$ = 30.2 years), $^{239}$Pu ($t_{1/2}$ = 24,110 years), and $^{240}$Pu ($t_{1/2}$ = 6,563 years). And among these, $^{137}$Cs, $^{239}$Pu, and $^{240}$Pu have become particularly valuable as tracers in soil erosion studies because of their trong association with fine soil and sediment particles.

The $^{137}$Cs artificial radionuclide, a fission product of plutonium and uranium, is one of the major byproducts of NWT and nuclear fuel burn-up, with a yield of more than 6% (Kurihara et al., 2020). Owing to its relatively short half-life, $^{137}$Cs is highly radioactive and is a source of both beta minus and gamma particles. The plutonium isotope $^{239}$Pu is either part of the fissile material in nuclear weapons and reprocessed nuclear fuels or is formed when $^{238}$U interacts with a neutron. The heavier plutonium isotope $^{240}$Pu, considered an impurity in the fissile material of nuclear weapons (Şahin, 1981), is subsequently generated from $^{239}$Pu through neutron capture as well. Both isotopes decay by alpha emission and were tradionally measured by alpha spectrometry, which cannot effectively discriminate between their energies. Hence, $^{239}$Pu and $^{240}$Pu have been reported together as "$^{239+240}$Pu", and this nomenclature has likewise been applied throughout the paper.

While radioactive debris from NPP accidents were spread in the lower atmosphere, radioactive debris from the NWT were introduced into the atmosphere at various heights depending on the type of nuclear testing conducted (e.g., barge/ship, balloon, airdrop, airburst), the explosive yield of the bomb, and the weather conditions as well as the amount and distribution of rainfall during re-deposition. As opposed to NPP accidents, $^{137}$Cs and $^{239+240}$Pu from NWT are deposited more uniformly due to temperatures reaching far beyond the temperature of volatilization for Cs and Pu (Brode, 1964; Meusburger et al., 2020; Kirchner er al., 2011; Steinhauser et al., 2014) as well as the longer time frame in which differences in rainfall patterns are equalised. Upon explosion, radioactive substances rapidly attach to ambient aerosols and disperses according to airflow patterns (Bennett, 2002; Corcho

Alvarado et al., 2014). Aerosols with larger particle sizes > 50 μm are deposited immediately within a few hundred km, referred to as the "local fallout", and considered to have limited global implications (Garcia Agudo, 1998; Bouisset et al., 2018). Debris injected into the troposphere remains suspended for up to a few weeks within the latitudinal band of injection, whereas debris injected into the stratosphere remains circulating for up to a year (Bennett, 2002) or even longer for finer aerosol particles < 0.1 μm (Corcho Alvarado et al., 2014). These radioactive particles found their way to the ground or surface water through wet and dry fallout depositions.

Among the host of FRNs deposited globally, $^{137}$Cs has been the most commonly used tracer of soil erosion in the past (Walling et al., 1998, 2007; Mabit et al., 2013, 2014). However, more than 60 years after the fallout from the NWT that peaked in the 1960s, $^{137}$Cs has undergone two half-lives and is now increasingly depleted and difficult to detect in many areas with relatively lower amounts of deposition, such as the Southern Hemisphere. Its measurement now also increasingly requires the use of low-background analytical facilities. In addition, the heterogeneous inputs from NPP accidents in the Northern Hemisphere, such as those of Chernobyl resulted in highly variable $^{137}$Cs inventories across Europe (Meusburger et al., 2020), especially across the Alps (Alewell et al., 2014). The latter is partly caused by the greater part of the Chernobyl deposition resulting from a few temporally and spatially very heterogeneous rainfall events. The $^{137}$Cs deposition on the partly snow-covered ground also resulted in heterogeneous and concentrated flow patterns during snow melt.

Owing to their longer half-lives, $^{239+240}$Pu have been increasingly recognized as an alternative tracer and chronological marker to assess soil erosion and/or the chronology of sediment deposition (Meusburger et al., 2023; Hancock et al., 2014; Alewell et al., 2017; Romanenko and Lujaniene, 2023). Globally, the spread of $^{239+240}$Pu is also less affected by NPP accidents such as the Chernobyl and Fukushima accidents, which deposited $^{239+240}$Pu mainly in confined proximal areas in Europe and up to ~200 km from the NPP site in Japan, respectively (Alewell et al., 2017). Although traces of $^{239+240}$Pu (and other actinides) derived from Chernobyl were found in areas as far as Scandinavia and in the Baltic Sea, they remained minor as compared to the contribution from the NWT (Lin et al. 2021; Salminen-Paatero et al., 2020). Since the atmospheric NWT were also conducted throughout the year over several decades, this means that $^{239+240}$Pu deposition due to NWT was more or less continuous, reducing the heterogeneity caused by deposition on snow-covered ground or the impact of a few heavy rainfall events.

Upon deposition on land, evidence suggests that $^{137}$Cs, similar to other monovalent ions, is rapidly and strongly adsorbed in the cation exchange sites of clay minerals to balance the negative charge on the alumino-silicate structure (Cornell, 1993; Mukai et al., 2016). $^{239+240}$Pu is (almost) irreversibly sorbed onto Fe/Mn oxides and/or forms complexes with organic matter (Kersting 2013; Lujianiene et al., 2002), in addition to its adsorption onto clay particles. However, the exact sorption mechanisms and differences between $^{137}$Cs and $^{239+240}$Pu may be more complex and likely overlap depending on the deposition environment.

While the occurrence of $^{137}$Cs and $^{239+240}$Pu in the environment continues to be monitored globally, especially in areas with localized exposure to nuclear accidents, these FRNs have gained interest as powerful tools for investigating critical Earth surface processes because of their close association with the soil particles (Alewell et al., 2017; Mabit et al., 2013). Their deposition coinciding with the onset of the Great Acceleration also provides an opportunity to study the widespread land degradation occurring since then and continuing nowadays. However,

our current scientific knowledge on the fallout chronology is better constrained in the Northern Hemisphere, whereas very little is known regarding the timing and the spatial distribution of their deposition in the Southern Hemisphere (Foucher et al., 2021).

## 1.2 Objectives

The aim of this review and meta-analysis is to update our current knowledge on the distribution of fallout $^{137}$Cs and $^{239+240}$Pu in Southern Hemisphere soils as part of the Franco-Swiss funded AVATAR Project (https://avatar-project.net/). The main objective of the AVATAR Project is to understand better the fallout chronology and distribution of $^{137}$Cs and $^{239+240}$Pu in the Southern Hemisphere for various environmental applications. Here, we synthesized the history and origins of $^{137}$Cs and $^{239+240}$Pu in the Southern Hemisphere and identified gaps in the reported data. We then compiled reference soil $^{137}$Cs and $^{239+240}$Pu data from the literature to build a database. Using data from the literature, we determined the distribution of $^{137}$Cs and $^{239+240}$Pu and their possible sources using their isotopic ratios. Finally, we demonstrated in a case study how the database can be utilized to identify which environmental factors, such as climate, topography, and geographic location, affect the distribution of $^{137}$Cs and $^{239+240}$Pu in reference soils.

## 2 Origins of fallout $^{137}$Cs and $^{239+240}$Pu in Southern Hemisphere soils

In contrast to the Northern Hemisphere, where the Chernobyl or Fukushima NPP accidents influenced FRN deposition to a crucial extent in many regions, virtually all FRNs deposited in the Southern Hemisphere soils originate from the past atmospheric NWT carried out by different nuclear states (Fig. 1). The first NWT occurred in 1945 with the Trinity Test atomic bomb testing in New Mexico by the United States, which yielded a total of 21 kt of energy and resulted in an 11 kt injection into the local/regional atmosphere and a 10 kt injection into the troposphere (UNSCEAR, 2000). As this relatively low-yield test was carried out atop a 30m tower above 30° N, it can be assumed that none of the radioactive debris reached the Southern Hemisphere soils via wet and/or dry deposition. The 1945 bombings in Hiroshima and Nagasaki that followed after the Trinity Test with yields of 21kt and 15 kt, respectively, were detonated at a higher altitude at 500-600 m. However, the yields were relatively minimal causing the fallout to be more localized (Saito-Kobuu et al., 2007), and could not have significantly affected the Southern Hemisphere. Most NWTs that also occurred shortly after these were performed in the Northern Hemisphere until 1951 (Fig. 1).

Atomic and thermonuclear bombs differ fundamentally in their design. Atomic bombs exclusively utilize energy from the fission of fissile materials, which is primarily made up of $^{239}$Pu or $^{235}$U. The more powerful thermonuclear weapons obtain their yield from the fission of $^{239}$Pu or $^{235}$U in a primary stage, which then initiates fusion reactions of deuterium or tritium fuel in a secondary stage. Thermonuclear weapons contributed over 90% of the radioactive debris from atmospheric NWT (UNSCEAR, 2000).

With the development of high-yield thermonuclear bombs in the 1950s, the majority of the radioactive debris from the NWT was injected into the stratosphere and high equatorial atmosphere equivalent to 139 Mt and 6.36

160    Mt of fission energy, respectively (UNSCEAR, 2000). Upon settlement and mixing in the lower equatorial stratosphere, eddy diffusion allowed for the inter-hemispheric mixing of radioactive debris before deposition. The yield injected into the stratosphere and high equatorial atmosphere is much higher than the 15.6 Mt equivalent fission energy injected into the troposhere, the debris of which remained more confined within a certain latitudinal band (UNSCEAR, 2000). The first high-yield thermonuclear weapon test, codenamed Ivy Mike, was performed

at Enewetak Atoll (11.55° N, 162.31° S) in November 1952 with a yield of 10.4 Mt (UNSCEAR, 2000). While this test was carried out in the Northern Hemisphere, its proximity to the equator and its high yield resulted in FRN deposition in the Southern Hemisphere. Accordingly, radioactive fallout in the Southern Hemisphere likely started only after the Ivy Mike test in 1952, as was later on observed in Antarctic ice cores by Koide et al. (1985).

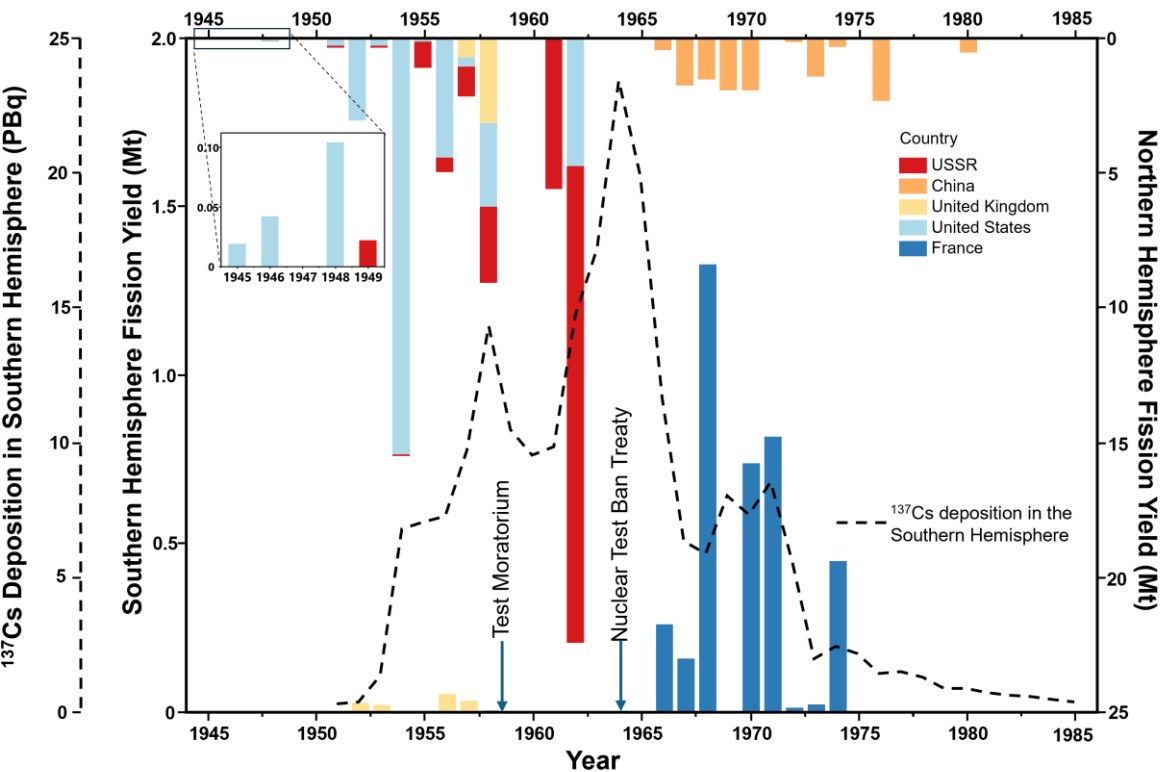

**Figure 1. Fission yields of atmospheric nuclear weapons testing in the Northern and Southern Hemispheres (UNSCEAR, 2000) and ¹³⁷Cs deposition in the Southern Hemisphere (broken lines), which peaked in 1964 at 23 PBq. Due to the discontinuous monitoring of the United States Atomic Energy Commission's Environmental Measurements Laboratory between 1954 and 1976, 40-50% of the data are missing (Evrard**

**et al., 2020), casting a doubt on the fallout in the Southern Hemisphere. (Note: Some very low-yield tests are not visible in the graph, including the French tests conducted in Algeria. For a complete list, see *Annex C* in UNSCEAR (2000). Axes for fission yields are also scaled differently for emphasis.)**

Fission reactions produce most of the FRNs as byproducts, including $^{137}$Cs. In addition, the yield from fission also provides an estimate of how much fissile material ($^{239}$Pu or $^{235}$U) was used in the testing. On the other hand, fusion reactions mainly produce $^3$H, $^{14}$C, $^{54}$Mn, and $^{55}$Fe as FRNs (UNSCEAR, 2000) in addition to $^4$He, neutrons, and large amounts of energy. Between 1945 and 1980, an estimated total of 502 atmospheric tests with a cumulative yield of 440 Mt were conducted globally, and more than 90% of these tests were based in the Northern Hemisphere (UNSCEAR, 2000). Of the 440 Mt yield released, 189 Mt originated from fission (UNSCEAR, 2000), the partition which is relevant in determining the amount of $^{137}$Cs and $^{239+240}$Pu released after a test. In the Northern Hemisphere, the majority of the fission yield was generated by the NWT conducted by the USA and USSR. In contrast, those in the Southern Hemisphere, albeit much lower in terms of yield, were dominated by France (Fig. 1).

**2.1 Gaps in reported data**

According to the global monitoring network operated by the United States Atomic Energy Commission's Environmental Measurements Laboratory (EML), approximately 23.8% of the fallout from all past atmospheric NWT was deposited in the Southern Hemisphere (UNSCEAR, 1982, UNSCEAR, 2000). This proportion was estimated from $^{90}$Sr measurements in air filters, which served as a proxy for other FRNs such as $^{137}$Cs and $^{239+240}$Pu (*see Annex C in* UNSCEAR, 2000). However, as opposed to $^{137}$Cs, the origin of fallout $^{239+240}$Pu differs from that of $^{90}$Sr. $^{90}$Sr and $^{137}$Cs are generated only in a series of fission chains, whereas $^{239+240}$Pu are already part of the fissile material in nuclear weapons. Thus, using $^{90}$Sr to estimate fallout $^{239+240}$Pu can be expected to lead to inaccuracies in these calculations/reconstructions. This is especially true for the Southern Hemisphere, where fallout sources differ depending on location.

The deposition of FRNs in the Southern Hemisphere supposedly peaked in 1964-1965, based on the EML data, as reported by UNSCEAR (2000). However, 50% of the data from EML monitoring stations were missing for 1954-1976 (Evrard et al., 2020) as most of the monitoring stations did not have continuous records for this period (Chaboche et al., 2021; Hardy, 1977). Thus, EML may not have accurately accounted the fallout from the French NWT, whose emissions peaked in 1968. In addition, since most stations only began recording measurements between 1956 and 1958 (Hardy, 1977), the fallout from the British NWT in Australia, which started in 1952 and 1953, may not have been considered. Therefore, the UNSCEAR may have misestimated the fallout distribution in the Southern Hemisphere, especially in areas affected by the French and British fallouts. A significant fraction of the Southern Hemisphere is also covered with oceans exposed to different precipitation regimes than terrestrial environments, and the fallout in these regions has not been monitored. These gaps in data for the Southern Hemisphere highlight the need to review and reevaluate the fallout in the Southern Hemisphere to refine the methods that rely on FRN data, especially those of $^{137}$Cs and $^{239+240}$Pu.

## 3  Literature survey and analytical approaches

**3.1 AVATAR-Soils Database: A Database of $^{137}$Cs and $^{239+240}$Pu in Equatorial and Southern Hemisphere Reference Soils**

Using the Thomson Reuters Web of Science platform, we conducted a literature survey until October 2024 to build the AVATAR-Soils Database (https://doi.org/10.5281/zenodo.14008220). The search keywords "soil", "cesium", "137Cs", "Cs 137", "plutonium", "239+240Pu", "Pu 239" and "Pu 240" were used in isolation and/or combination with the names of the countries found in the Southern Hemisphere. All countries in South America and sub-Saharan Africa were included in the current literature survey to consider the NWT conducted near the

equator; to prevent discontinuity in continents whose areas cross the equator, and to consider the seasonal movement of the intertropical convergence zone (ITCZ) around the equator. Publications in Portuguese and Spanish languages and PhD dissertations that reported on soil $^{137}$Cs and $^{239+240}$Pu were also included in the selection.

For a soil profile to be included in the database, the following two conditions had to be satisfied: (i) collection

from an undisturbed area, or the so-called "reference" site, located in a flat landscape that has, to the best possible assessment, not been affected by soil redistribution processes such as erosion and/or deposition in recent decades or since the fallout period (Arata et al., 2017; Kirchner, 2013); and (ii) the sampling details and site characteristics were provided.

Reported reference soils that were either (a) flooded, (b) drained, (c) repeated from articles and/or applications

already considered, (d) located on a slope; (e) located on a farm where cultivation has been implemented; and/or (f) the sampling locations of which were not provided or could not be obtained were excluded from the AVATAR-Soils Database.

### 3.2 Statistical and modelling approaches

#### 3.2.1 Decay-correction for $^{137}$Cs

To compare $^{137}$Cs data spanning years or decades between measurements, all $^{137}$Cs data were decay-corrected to 2024 with the following equation:

$$^{137}Cs_{2024} = {}^{137}Cs_{literature}\,e^{-\lambda t} \tag{1}$$

where λ is the decay constant of $^{137}$Cs (λ = ln2/30.2 y) and t is the time in years since the sampling year. For $^{137}$Cs data, for which neither dates of sampling nor of decay-correction were provided, we assumed a 4-year delay

between the day of sampling and publication. This corresponds to the mean time delay between sampling and publication in the current literature survey and was calculated from articles in which both dates were provided. The same approach was applied by Chaboche et al. (2021) and Jagercikova et al. (2015) to decay-correct $^{137}$Cs data for which the associated dates of sampling and decay-correction were not available. To calculate the mean time delay in this study, we excluded data from a resampling study that spanned over decades (Loughran and

Balog, 2006) from the calculation.

#### 3.2.2 Latitudinal distribution comparisons

To compare the latitudinal distributions of $^{137}$Cs and $^{239+240}$Pu inventories, inventory data were bootstrapped in R for 10,000 iterations (R Core Team, 2024). Bootstrapping is a resampling and replacement technique that allows

inference from sample data without making strong distributional assumptions (Haukoos and Lewis, 2005; Mooney and Duval, 1993). It is therefore appropriate to compare datasets with large variations, such as the [137]Cs and [239+240]Pu inventories available from the AVATAR-Soils Database. The bootstrapped means, reported with 95% confidence interval, were compared with previously reported global data analysed from air filters ([137]Cs derived from [90]Sr) and soil samples ([239+240]Pu) by the EML (UNSCEAR, 2000; Hardy et al., 1973).

### 3.2.3 Two-source unmixing model

The isotopic ratios of FRN in the soil are affected by the isotopic signatures of the fallout sources (*see Sect. 5.2*). To un-mix these sources, we used an un-mixing model previously used to evaluate the contribution of fallout from two sources, where both the global fallout and local/regional fallout have occurred at a given site (Kelley et al., 1999; Bouisset et al., 2021; Chaboche et al., 2022).

Isotopic ratios originating from low-yield tests such as the French and British NWT result in low [240]Pu/[239]Pu atom and [137]Cs/[239+240]Pu activity ratios. As such, it is impossible to separate French from British fallout based on these isotopic ratios. An assignment to either British or French NWT sources can be done depending on the location of the fallout origin, as it is known that in the Southern Hemisphere, the French tests were performed in French Polynesia. In contrast, the British tests were conducted in Australia and the Malden Islands. However, the British testing in the Malden Islands was far enough from other Equatorial and Southern Hemisphere land areas, with a wind direction towards the equator (Chamizo et al., 2020), to have a significant fallout contribution in this part of the world.

When French or British fallout is mixed with the global fallout, the ratio $R_s$ in the soil may, therefore, be expressed by the following equation:

$$R_s = x_{B/FF} \times R_{B/FF} + \left(1 - x_{B/FF}\right) \times R_{GF} \tag{2}$$

where $R_{B/FF}$ is the ratio of British or French fallout, $R_{GF}$ is the ratio of the global fallout, $x_{B/FF}$ is the relative contribution of the British or French fallout, and $\left(1 - x_{B/FF}\right)$ is the relative contribution of the global fallout. The relative contribution of the British or French fallout may, therefore, be calculated via the following equation:

$$x_{B/FF} = \frac{R_s - R_{GF}}{R_{B/FF} - R_{GF}} \tag{3}$$

The standard uncertainty of the contribution $u(x_{B/FF})$ was determined by the combination of uncertainties associated with each variable in Eq. (3), and was expressed as follows:

$$u(x_{B/FF}) = x_{B/FF} \times \sqrt{\left(\frac{u(R_s)}{R_s - R_{GF}}\right)^2 + \left(\frac{u(R_{B/FF})}{R_{B/FF} - R_{GF}}\right)^2 + \left(\frac{(R_s - R_{B/FF}) \times u(R_{GF})}{(R_s - R_{GF}) \times (R_{B/FF} - R_{GF})}\right)^2} \tag{4}$$

### 3.2.4. Assessing inventory predictability in a case study with Random Forest

Since FRNs have increasingly become an indispensable tool in many fields of scientific research, such as environmental tracing and geomorphological studies, it is critical to obtain data in areas lacking data. One of the important aims of the AVATAR-Soils Database is the prediction of baseline $^{137}$Cs and $^{239+240}$Pu inventories in areas devoid of data. To achieve this goal, the predictability of the inventories in the current database has to be tested using possible explanatory variables or covariates.

Accordingly, geospatial and climatic variables, including rainfall, are considered the most important predictors of $^{137}$Cs and $^{239+240}$Pu deposition in the soil as determined in previous studies (i.e., Chappel et al., 2011, Meusburger et al., 2020, Chaboche et al., 2021). We, therefore, used geospatial and historical bioclimatic variables (Table S1) as potential covariates in this case study. The historical bioclimatic data were extracted from WorldClim 2.1 climate data for 1970-2000 (Fick and Hijmans, 2017) at a 30 arc-second (~ 1km) spatial resolution. A total of 22 potential covariates were used in this analysis, as detailed in Table S1. The correlations among and between these covariates and the inventories were tested via Spearman's rank correlation (Fig. S1).

A Random Forest algorithm (Breiman, 2001) was run in R (R Core Team, 2024) via the *ranger* package (Wright and Zegler, 2015) to determine the best covariates that explain the variations in the $^{137}$Cs and $^{239+240}$Pu inventories while also showing the application of the AVATAR-Soils database. The selection of the Random Forest model is based on existing studies that consistently use it as their first choice among various machine-learning models (Hong et al., 2024; Shuryak, 2022; Gupta et al., 2022). The optimal value for the most sensitive hyper-parameter, 'mtry,' in the Random Forest model was determined using five-fold cross-validation. Default settings from the ranger package were used for the remaining hyper-parameters, such as the number of trees, minimum node size, maximum tree depth, and splitting rule. In the cross-validation process, the data were randomly divided into five parts, each containing 20% of the total data. The Random Forest model was trained five times. Each time, one of the parts was used as a validation set, while the others were used for training. The validation results were then combined and compared with the measured data to assess model accuracy. Model accuracy was evaluated via the root mean square error (RMSE), coefficient of determination ($R^2$), and concordance correlation coefficient (CCC) (Lawrence, 1989). This process was repeated for each 'mtry' value to find the optimal value, and the entire cross-validation procedure was repeated three times to ensure robustness. The final correlation plot was created using the optimal 'mtry' and averaged predictions from the three cross-validation repetitions. Additionally, the relative importance of each variable was evaluated via the residual sum of squares (RSS) metric (Gupta et al., 2021). A lower RSS indicates a more important covariate, with the second-lowest RSS identifying the second-most important covariate, and so on.

## 4 AVATAR-Soils Database overview

### 4.1 Publication distribution, trends, and applications

From the 1526 publications screened, only a total of 123 publications reporting 1122 reference soil profiles with $^{137}$Cs and $^{239+240}$Pu data were included in the database. Among these soil profiles, 999 (89.0%) had $^{137}$Cs data, and 123 (11.0%) had $^{239+240}$Pu data, but only 29 (2.6%) had both $^{137}$Cs and $^{239+240}$Pu data.

The earliest publication in the database on $^{137}$Cs in Equatorial and Southern Hemisphere reference soils was published by Loughran et al. (1988), who estimated soil erosion in a drainage basin in Hunter Valley, New South Wales, Australia. Earlier publications on soil $^{137}$Cs profiles, such as those of Loughran et al. (1982) and Campbell et al. (1983), were available, but the geographic locations of the investigated profiles were difficult to determine. Accordingly, they were excluded from the database. While publications on $^{137}$Cs continued to increase in the following decades, the number of publications on $^{239+240}$Pu increased only after the 2010s (Fig. 2a). For $^{239+240}$Pu, the earliest publication in the database was that conducted by Hardy et al. (1973), which was based on the report released on the global inventory and the distribution of $^{238}$Pu from SNAP-9A based on EML data (Hardy et al., 1972). While this was the first publication that aimed to reconstruct a regional-scale baseline fallout inventory of plutonium in the Southern Hemisphere, a subsequent study using the same set of samples was conducted by Kelley et al. (1999).

Most reference soil profiles across the Equatorial and Southern Hemisphere land surfaces with $^{137}$Cs data were located in South America (69.4%). In comparison, Oceania (60.2%) has the greatest number of $^{239+240}$Pu measurements, mainly from Lal et al. (2020) and Hardy et al. (1973) (Fig. 2b). Despite sub-Saharan Africa covering a large portion of the Equatorial and Southern Hemisphere land area, measurements of $^{137}$Cs and $^{239+240}$Pu in the region represent only 12.2% and 17.1% of all available data, respectively.

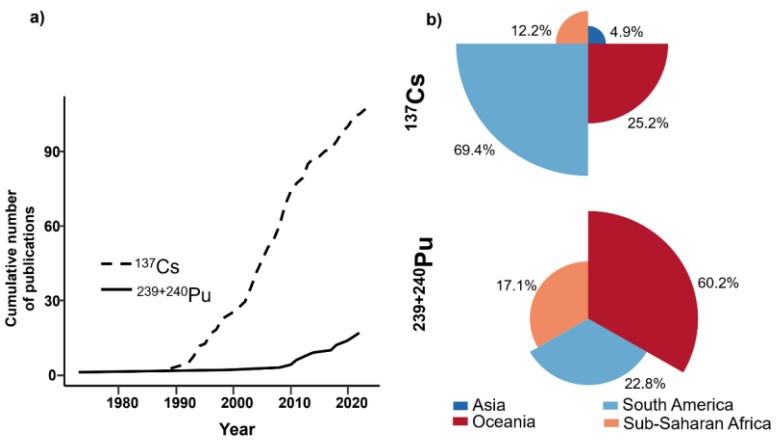

**Figure 2. (a) Cumulative number of publications on $^{137}$Cs (n=113) and $^{239+240}$Pu (n=24) in Equatorial and Southern Hemisphere reference soils and (b) location of reference soil profiles with $^{137}$Cs (n=999) and $^{239+240}$Pu (n=123) measurements across Equatorial and Southern Hemisphere continents.**

In terms of the applications for $^{137}$Cs and $^{239+240}$Pu measurements in reference soils, articles related to soil erosion assessment composed the majority of the publications, covering 68% (n=84) of the total number. This was followed by applications related to environmental radioactivity with 35% (n=43), and applications for sediment tracing and fingerprinting with the lowest number of publications (6%, n=7). For $^{137}$Cs publications, applications for soil erosion continued to increase until 2019 (Fig. 3), which implies the increased use of radionuclides for erosion assessment. While applications for environmental radioactivity assessment remain important for $^{137}$Cs measurements in reference soils, they began to decrease after 2009. Environmental radioactivity publications in Equatorial and Southern Hemisphere soils were driven primarily by the monitoring conducted by Labreque et al.,

(e.g., LaBrecque et al., 1993, LaBreque et al., 2007) of the Venezuelan National Science and Technology Foundation (CONICET) following the Chernobyl nuclear power plant accident, and by Schuller et al., (e.g., Schuler et al., 1996, Schuller et al., 1997) across Chile after recording seemingly higher $^{137}$Cs inventories than those previously estimated from the global weapons fallout. Despite $^{137}$Cs being used by the sediment tracing and fingerprinting community (Evrard et al., 2020), $^{137}$Cs measurements in reference soils for this application are uncommon. The $^{137}$Cs data in source soils that have been eroded are more commonly reported alone, especially for large catchments where on-site erosion rates do not necessarily translate to the percent contribution of sediments in lakes, dams, and reservoirs because they can be stored in channels (Wallbrink et al., 1998). For example, only 10% of the studies (n=30) in Equatorial and Southern Hemisphere countries reviewed by Evrard et al. (2020) reported $^{137}$Cs measurements in reference soils.

For the $^{239+240}$Pu publications, studies in reference soils prior to 2000 were only conducted for environmental radioactivity assessment (Fig. 3). Everett et al. (2008) were the first to measure $^{239+240}$Pu in Equatorial and Southern Hemisphere reference soils for both soil erosion and sediment tracing applications, comparing its concentration with that of $^{137}$Cs in the Herbert River catchment in Australia and its suitability as an alternative to $^{137}$Cs. The suitability of using $^{239+240}$Pu as a tracer for soils and sediments, as well as the development and improvement of measurement techniques, has driven the increase in the use of $^{239+240}$Pu as a tracer for soil erosion (*for an overview, see* Alewell et al., 2017) and sediment transport (Romanenko and Lujaniene, 2023).

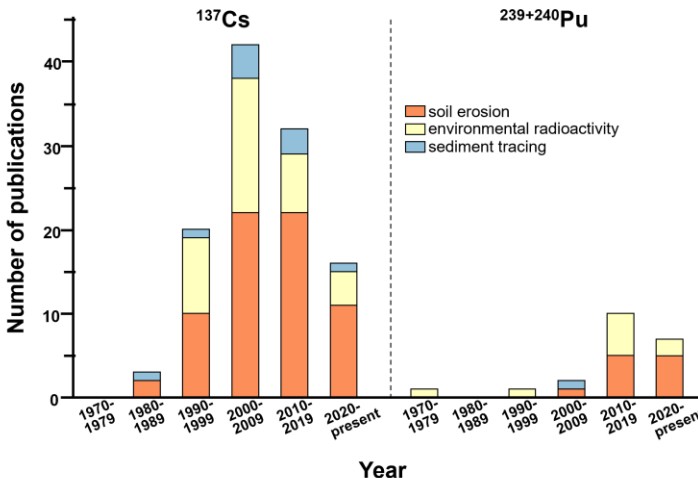

**Figure 3. Temporal trends of publications using $^{137}$Cs and $^{239+240}$Pu data in reference soils for different applications.**

### 4.2 $^{137}$Cs and $^{239+240}$Pu data availability in the literature

Among the 999 soil profiles analysed for $^{137}$Cs, 429 (42.9%) had inventory data, 160 (16.0%) had both inventory and activity data, 269 (26.9%) had inventory data only, and 570 (57.0%) had activity data only. Among the soil profiles with $^{137}$Cs data, 297 (29.7%) were decay-corrected to a particular date, 475 (47.5%) had sampling dates

provided, and 643 (64.3%) had either the date of decay correction or the date of sampling. However, in 356 (35.6%) of the published $^{137}$Cs data, neither the date of decay correction nor the date of sampling was recorded.

For the 123 soil profiles analysed for $^{239+240}$Pu, owing to the differences in the analytical techniques used, with techniques other than alpha spectrometry (i.e., ICP-MS, TIMS and AMS) being able to measure the two isotopes separately, some activity and inventory data had to be derived from the reported individual isotopes. In total, 102 (82.6%) soil profiles had inventory data, 28 (22.8%) had both inventory and activity data, 73 (71.6%) had inventory data only, and 18 (14.6%) had activity data only. The $^{240}$Pu/$^{239}$Pu atom ratio was either reported or derived for 91 (74.0%) soil profiles. The $^{137}$Cs/$^{239+240}$Pu activity ratios were provided or derived for 24 (19.5%) reference soil profiles.

## 5   Distribution and sources of fallout $^{137}$Cs and $^{239+240}$Pu inventories

### 5.1  Distribution of $^{137}$Cs and $^{239+240}$Pu inventories

The distributions of published $^{137}$Cs and $^{239+240}$Pu inventories in Equatorial and Southern Hemisphere soils are located mostly along the edges of continents (Fig. 4). In contrast, little is known about inventories inside of continents such as in northwestern Brazil or Bolivia in South America, in the Democratic Republic of Congo, Angola, Namibia, and Botswana in Sub-Saharan Africa, and in the arid regions of central Australia.

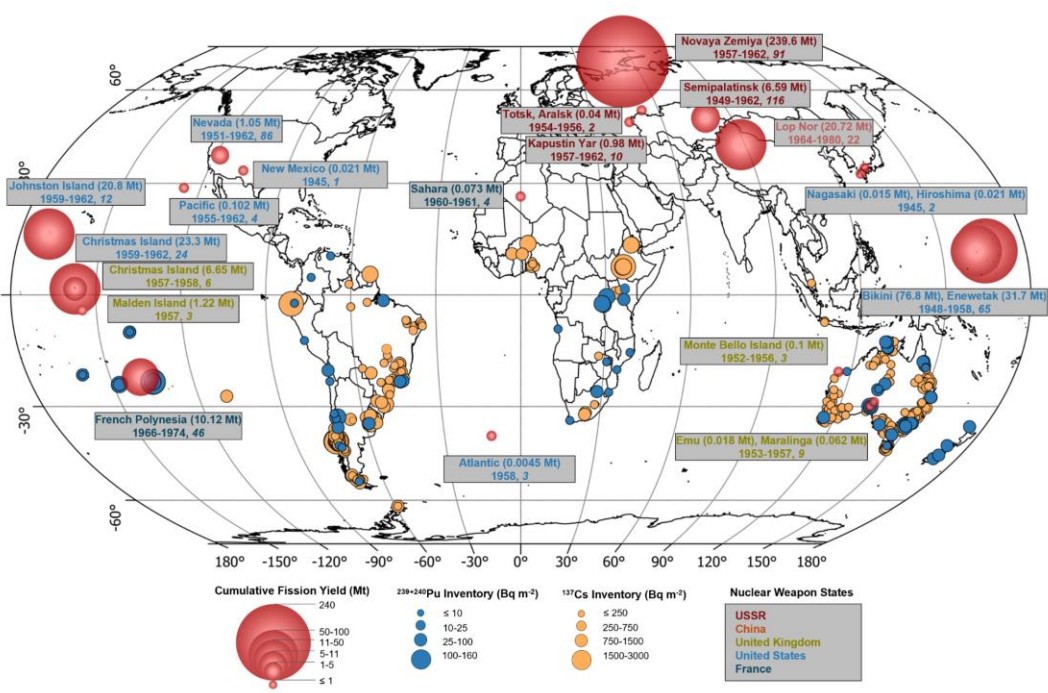

**Figure 4. Cumulative yield of global atmospheric nuclear weapon detonations (modified from Chaboche et al., 2022) and FRN ($^{137}$Cs and $^{239+240}$Pu) inventories in equatorial and Southern Hemisphere reference soils. Does not include Vixen B series in Australia which used 22 kg plutonium (Johansen et al., 2014). $^{137}$Cs inventories were decay-corrected to 2024.**

Since most of the NWTs occurred in the Northern Hemisphere, high $^{137}$Cs inventories were recorded above the equator (Fig. 5a; Table S2), mostly in Sub-Saharan Africa. In the Southern Hemisphere latitudinal bands, relatively high $^{137}$Cs inventories were recorded in South America, while lower $^{137}$Cs inventories were recorded in Asia and Oceania, excluding Polynesia, which was at least partly directly and significantly affected by local fallout from the French NWT (Bouisset et al., 2021). For $^{239+240}$Pu inventories (Fig. 5b; Table S2), high values were also recorded in Sub-Saharan Africa closest to the equator, within the 0-10° N latitudinal band. For South America, high $^{239+240}$Pu inventories were observed in the 30-40° S latitudinal band, and Polynesia had the highest inventories recorded, most likely because of its proximity to the French NWT grounds.

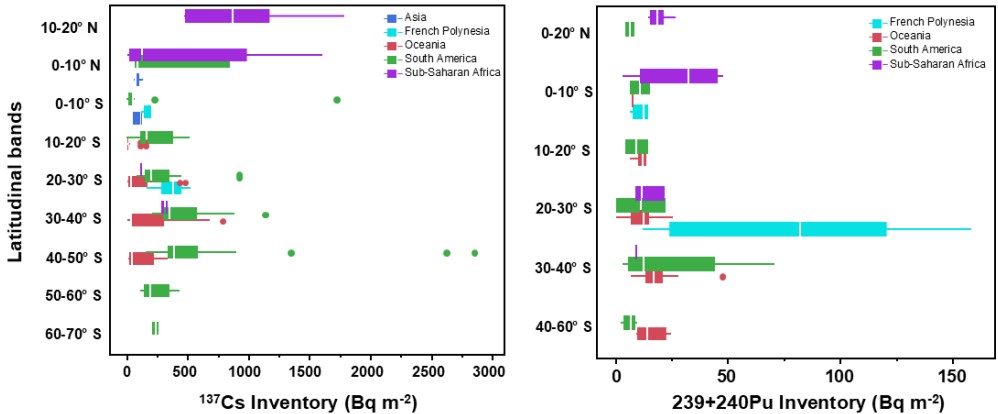

**Figure 5. Boxplots of the latitudinal distribution of $^{137}$Cs and $^{239+240}$Pu inventories reference soils, classified per continent. Inventories from French Polynesia are grouped together. Mean, median, 25$^{th}$ percentile, and 75$^{th}$ percentile values are presented in Table S2.**

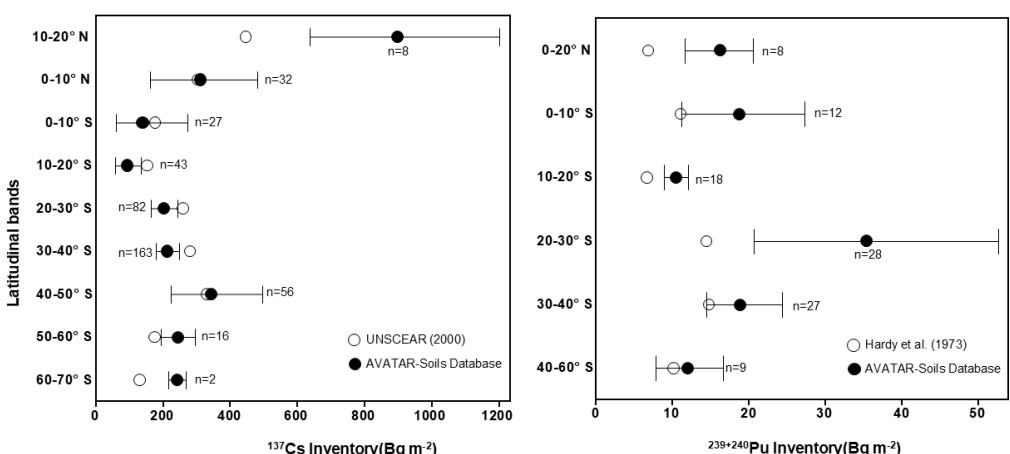

**Figure 6. Latitudinal band comparisons of $^{137}$Cs and $^{239+240}$Pu inventories between the AVATAR-Soils Database and EML as reported by UNSCEAR (2000) for $^{137}$Cs and Hardy et al., (1973) for $^{239+240}$Pu. Error bars are the 95% confidence intervals of the means simulated using 10000 bootstrapped iterations in R. The numbers indicated correspond to the number of reference soil inventories in the AVATAR-Soils Database.**

Based on the inventories compiled for the entire Equatorial and Southern Hemisphere region in the AVATAR-Soils Database, the $^{137}$Cs inventories were highest north of the equator. In the Southern Hemisphere, $^{137}$Cs inventories started to increase from the 20-30° S latitudinal band and peaked in the 40-60° S latitudinal bands (Fig. 6a). Furthermore, the $^{239+240}$Pu inventories were the highest within the 0-10° and 20-30° S latitudinal bands, but the lowest above the equator at 0-20° N (Fig. 6b). One of the reasons for these differences in the latitudinal

distributions between the $^{137}$Cs and $^{239+240}$Pu inventories may be related to the differences in the locations of the sampled reference soils. For instance, in South America, the reference soils with $^{137}$Cs inventory data are mainly located on the eastern side of the continent, while those with $^{239+240}$Pu inventory data are mainly found more on the western side (Fig. 4). In Sub-Saharan Africa, reference soils with $^{137}$Cs inventories are mainly located in the northern part, whereas those with $^{239+240}$Pu inventory data are mainly found more in the southern part. However,

part of these differences in distribution could also be a result of the different $^{137}$Cs/$^{239+240}$Pu activity ratios of the fallout sources (*see Sect. 5.2*).

### 5.1.1 Comparison with previous fallout reconstructions

Fallout reconstructions based on actual reference soils have been conducted before to evaluate the global fallout distribution of 137Cs inventories proposed by UNSCEAR (2000). At a global scale, previous work to reconstruct

the fallout distribution from the literature was conducted by Aoyama et al. (2006). Regional scale reconstructions of $^{137}$Cs inventories in the Southern Hemisphere were previously conducted by Chaboche et al. (2021) for South America and Chappel et al. (2011) for Australia. For $^{239+240}$Pu, Hardy et al. (1973) reported global $^{239+240}$Pu baseline data from samples collected from 65 sites around the world by the EML, with 30 sites located in the Southern Hemisphere. Aliquots from 54 samples from these sites and two additional ones were reanalyzed by

Kelley et al. (1999) to compare the baseline data generated with those obtained with novel analytical techniques.

The data compiled by Aoyama et al. (2006) suggest that at the global scale, UNSCEAR (2000) underestimated the total fallout in the entire Northern Hemisphere down to 15° S but overestimated the total fallout within the 25-35° S latitudinal band (Aoyama et al., 2006), with no data available for further latitudinal bands. However, the dataset for the Southern Hemisphere in Aoyama et al. (2006) was based on only 7 publications and 66 soil profiles,

which is extremely limited for providing reliable estimates for all the land surface in this part of the world. Chaboche et al., (2021) also reported discrepancies between the fallout distribution proposed by UNSCEAR (2000) and the mean $^{137}$Cs inventories in undisturbed soils in South America from the literature for each 10° latitude band. On the basis of compiled data from published research, $^{137}$Cs inventories between 30° and 50° S were found to be higher than the proposed distribution by UNSCEAR (2000), while the $^{137}$Cs inventories in soils

were found to be lower between the 0° and 10° S latitudinal bands (Chaboche et al., 2021). Unlike Aoyama et al., (2006) and Chaboche et al. (2021), Chappel et al. (2011) reported that the latitudinal distribution proposed by UNSCEAR (2000) was in good agreement with the $^{137}$Cs inventories measured in reference soil samples across Australia.

A comparison of the means per latitudinal band between the $^{137}$Cs inventories compiled in the AVATAR-Soils

Database and the distribution proposed by UNSCEAR (2000) is shown in Fig. 6a. The $^{137}$Cs inventories generally agree in the equatorial region (10° N-10° S) and within the 50-50° S latitudinal bands. However, the distribution proposed by UNSCEAR (2000) is likely underestimated in the 10-20° N and 50-70° S latitudinal bands, but

overestimated in the 10-40° S latitudinal bands. Incidentally, the overestimated region are also among the regions

with the highest number of measurements reported in the literature. This implies that measurements done within these latitudinal bands during the period of discontinuous monitoring may have focused on areas with relatively higher fallout. On the other hand, the latitudinal bands with underestimated inventories from UNSCEAR (2000) have the lowest number of reference soil inventories. This indicates that more measurements may be needed to confirm whether the distribution proposed by UNSCEAR (2000) in these latitudinal bands are indeed inaccurate.

The means of the $^{239+240}$Pu inventories compiled in the AVATAR-Soils Database appear to be higher in most of

the Southern Hemisphere land surface (0-40° S; Fig. 6b) than in the earlier estimates provided by the EML (Hardy et al., 1973). As these comparisons are based on a dataset that remains limited, more measurements of $^{239+240}$Pu inventories are likely needed to better understand the distribution of fallout $^{239+240}$Pu in Equatorial and Southern Hemisphere reference soils.

### 5.2 Fallout sources

Although the past atmospheric NWT were mostly conducted in the Northern Hemisphere, a considerable part of their debris also deposited in the Southern Hemisphere because of their long residence time, long-range transport, and atmospheric mixing. Thus, one of the primary sources of fallout $^{137}$Cs and $^{239+240}$Pu in Southern Hemisphere soils include the NWT dominated by the USA and USSR, referred to as the "global fallout" (Table 1). This occurred in addition to those derived from the generally low-yield NWT conducted in the Southern Hemisphere,

such as the French NWT in French Polynesia referred to as the "French fallout" and those performed by the UK in Australia referred to as the "British fallout." A distinct signature of the earlier NWT dominated by the USA before the test moratorium was signed in 1958 was also documented in dated ice cores in the Southern Hemisphere (Koide et al., 1979; Koide et al., 1985) and is referred to as the "pre-moratorium fallout".

**Table 1. Summary of $^{239}$Pu/$^{240}$Pu atomic and $^{137}$Cs/$^{239+240}$Pu activity ratios reported from the literature. A compilation of the values reported for each publication are presented in Table S3.**

| Fallout source | Nuclear states responsible | Testing period | Measurement location | $^{240/239}$Pu atomic ratio | References | $^{137}$Cs/$^{239+240}$Pu activity ratio | References |
|---|---|---|---|---|---|---|---|
| Pre-moratorium | USA (more dominant) and USSR | 1945-1958 | Arctic | 0.24 ± 0.03 | Koide et al., 1985 | 11.88 | Koide et al., 1982 |
| | | | Antarctic | 0.29 ± 0.05 | Koide et al., 1985 | 10.76 ± 1.08 | Koide et al., 1979, 1982 |
| British | UK | 1952-1958 | Australia | 0.042 ± 0.015 | Child and Hotchkis 2013; Johansen et al., 2014, 2019; Tims et al., 2013 | 0.269 ± 0.088 | Tims et al., 2013; Johansen et al 2019 |
| Global Fallout | USSR (more dominant) and USA | 1958-1963 | Northern Hemisphere | 0.18 ± 0.004 | Kelley et al., 1999; Meusburger et al., 2016, 2018, 2020; Krey et al., 1976; McArthur and Miller 1989; Krey and Beck 1981 | 20.20 ± 1.96 | McArthur and Miller 1989; Meusburger et al., 2016; Krey and Beck 1981; Hodge et al., 1996; Hardy 1975; Kim et al., 1998; de Bortoli et al., 1968; HASL 1973 as cited in Bertine et al., 1983; Earkins et al., 1981 as cited in Hodge et al., 1996 |
| | | | Southern Hemisphere | 0.18 ± 0.008 | Kelley et al., 1999 | | |
| French Fallout | France | 1964-1974 | French Polynesia | 0.035 ± 0.015 | Hrnecek et al., 2005; Chiappini et al., 1999; Chiappini et al., 1996; Chiappini et al., 1998; Bouisset et al., 2021 | 1.74 ± 0.035 | Bouisset et al., 2021 |

### 5.2.1. Isotopic fingerprints of fallout sources

Weapons-grade plutonium generally contains more than 93% of $^{239}$Pu, which is equivalent to a $^{240}$Pu/$^{239}$Pu atom ratio below 0.07 (Warneke et al., 2002; Ketterer and Szechenyi, 2008; Jones, 2019). However, this atom ratio changes in the resulting fallout due to nuclear transformations upon fission and the capture of fast neutrons by $^{238}$U in fusion devices (Hancock et al., 2014). Thus, because of the different weapon types and designs used by the different nuclear weapon states, each fallout sources have all their own distinct isotopic signature (Table 1). Low-yield detonations result in fallout with low $^{240}$Pu/$^{239}$Pu atom ratios, wheras high-yield detonations characteristic of thermonuclear bombs result in high $^{240}$Pu/$^{239}$Pu atom ratios due to high neutron fluxes (Corcho-Alavarado et al., 2022; Buesseler et al., 1997; Lachner et al., 2010). Notably, these isotope fractionations do not differ depending on whether the FRNs are injected into the troposphere or the stratosphere (Bouisset et al., 2021).

Global fallout has a $^{240}$Pu/$^{239}$Pu atom ratio of ~0.18 as determined from soil samples collected from 1970-1971 from regions around the world that were not influenc`ed by plumes from low-yield NWT, which are not characteristic of those collected during this period (Kelley et al., 1999). Other researchers have reported similar values elsewhere in the Northern Hemisphere (Meusburger et al., 2016, 2018, 2020; Krey et al., 1976; McArthur and Miller, 1989; Krey and Beck, 1981). Since the global fallout $^{240}$Pu/$^{239}$Pu signature was measured in soils collected from 1970-1971, it can be assumed that these soils also contained the plutonium from the pre-moratorium fallout. However, the $^{240}$Pu/$^{239}$Pu atom ratio of ~0.18 is also in agreement with the measurements conducted on air filters collected for the period 1959-1970 (HASL 1973 as cited in Bertine et al., 1983 and Koide et al., 1985). These findings suggest that the pre-moratorium fallout did not significantly contribute to soil $^{239+240}$Pu inventories. Recent investigations of freshwater lakes in the Southern Hemisphere also suggest that the pre-moratorium fallout contribution is much lower than that of other fallout sources (Guillevic et al., *in prep*).

The French fallout signature is characterized by a much lower $^{240}$Pu/$^{239}$Pu atom ratio of ~0.035 on the basis of different measurements of samples collected from French Polynesia (IAEA 1998; Hrnecek et al., 2005; Chappini et al., 1996). It is important to note, however, that this signature is only representative of tests and safety trials, which resulted in local fallout at the Mururoa and Fangataufa NWT sites (Table S3). This signature is used as a reference because it is the only one published to date, and no known signatures are available for the 37 other balloon-based nuclear tests, including the thermonuclear ones. For the British fallout, similarly low $^{240}$Pu/$^{239}$Pu atom ratios of ~0.04 have also been determined on samples collected near the Australian testing sites (Child and Hotchkis, 2013; Johansen et al., 2014, 2019; Tims et al., 2013). These low $^{240}$Pu/$^{239}$Pu atom ratios are indeed characteristic of low-yield NWT which happened 15 years apart. Importantly, among the fallout source isotopic signatures, only the global fallout was supported by measurements of air filter samples (Table S3). Therefore, the ratios determined from the soil or sediment samples may have received minor contributions from sources other than the local source to which these ratios were attributed to. Nevertheless, these isotopic ratios still provide useful tools for estimating the sources of fallout $^{239+240}$Pu in different environmental compartments globally.

Another indicator of the origin of FRN fallout is the $^{137}$Cs to $^{239+240}$Pu activity ratio, assuming that $^{137}$Cs and $^{239+240}$Pu are sorbed tightly to soil particles even after several decades. Reference soils in Colorado, which were reported to have mainly received radionuclides from the global fallout (as opposed to other sources of contamination such as plutonium processing plant) had a $^{137}$Cs/$^{239+240}$Pu activity ratio of ~20 (decay-corrected to

2024 for the current study; Hodge et al., 1996; Price 1991). Similar ratios have been reported elsewhere in the USA (McArthur and Miller, 1989; Krey and Beck, 1981; Hodge et al., 1996; Hardy 1975), in Italy (de Bortoli et al., 1968), in Scotland (Earkins et al., 1981 as cited in Hodge et al., 1996), and South Korea (Kim et al., 1998). However, Meusburger et al. (2016) reported a considerably higher variation in the $^{137}Cs/^{239+240}Pu$ activity ratio in the Haean catchment adjacent to the demilitarized zone in South Korea (Table S3). This is due to the different adsorption behaviours of $^{239+240}Pu$ and $^{137}Cs$ in the soil, with $^{239+240}Pu$ migrating to deeper layers than $^{137}Cs$ (Meusburger et al., 2016). For the French fallout, only Bouisset et al. (2021) provided a $^{137}Cs/^{239+240}Pu$ activity ratio of ~1.7, which was calculated in the same way as their proposed $^{240}Pu/^{239}Pu$ atom ratio. Similarly, Tims et al. (2013) and Johansen et al. (2019) also reported low $^{137}Cs/^{239+240}Pu$ activity ratios of 0.17 and 0.32 near the Australian testing sites in Maralinga and Montebello Islands, respectively. However, due to the limited measurements of $^{137}Cs/^{239+240}Pu$ activity ratios other than those characterizing the global fallout, as well as the differences in migration patterns between $^{239+240}Pu$ and $^{137}Cs$ in the soil, these ratios must be used with caution. To determine the $^{137}Cs/^{239+240}Pu$ activity ratios of the different sources more accurately, we propose that more reliable matrices that preserve the original ratio must be used, taking radioactive decay into account. These matrices could include coral archives or undisturbed rain gauge lake sediments where the percentage of the global fallout is known.

Owing to the high uncertainty associated with the $^{137}Cs/^{239+240}Pu$ activity ratios, only the $^{240}Pu/^{239}Pu$ atom ratios were used to determine the fallout sources in the AVATAR-Soils Database. In addition, data on the $^{137}Cs/^{239+240}Pu$ activity ratios were only available for 14 reference soils, which were collected near the testing sites in French Polynesia and in Maralinga. This approach is extremely insufficient for determining the sources of $^{137}Cs$ to $^{239+240}Pu$ in the large parts of the Equatorial and Southern Hemisphere regions.

### 5.2.2 $^{240}Pu/^{239}Pu$ atom ratios of reference soils

As shown in Fig. 7, the $^{240}Pu/^{239}Pu$ atom ratios of the French and British fallout overlap with each other. Although it has been shown that the French fallout is responsible for the shift in $^{240}Pu/^{239}Pu$ atom ratios from the global fallout in South America (Chaboche et al., 2022), it is not known whether it significantly contributed to the shifts of $^{240}Pu/^{239}Pu$ atom ratios in other regions. It was therefore assumed that contributions from the French fallout caused deviations in $^{240}Pu/^{239}Pu$ atom ratios from the global fallout in South America and the French Polynesia (e.g., Chaboche et al., 2022; Bouisset et al., 2021), while the British fallout caused similar deviations in the rest of Oceania (Froelich et al., 2019; Tims et al., 2013).

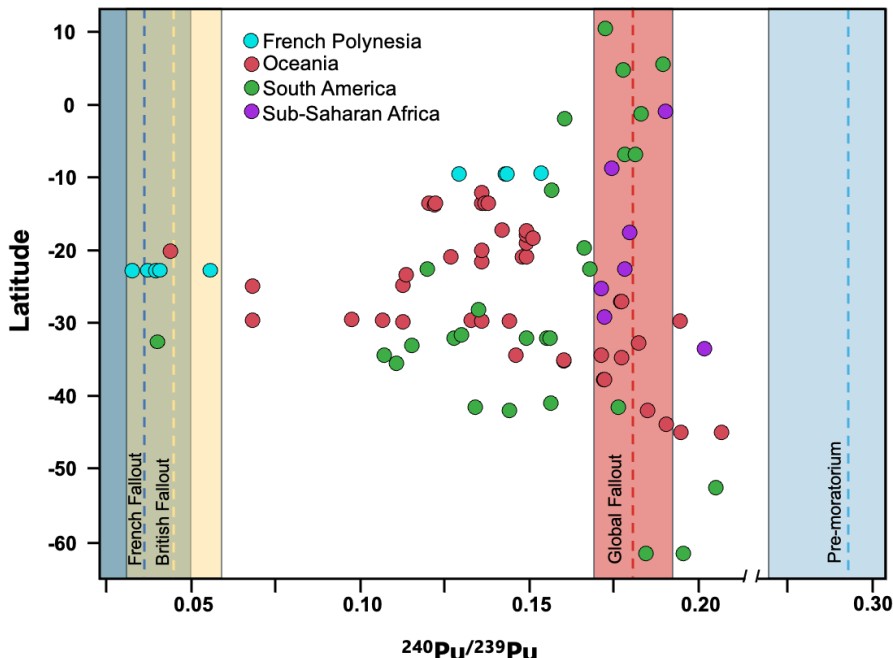

**Figure 7. Latitudinal distribution of $^{240}$Pu/$^{239}$Pu atom ratios in reference soils in comparison of the known fallout source ratios.**

The $^{240}$Pu/$^{239}$Pu atom ratios near the equator and below 45°S in the AVATAR-Soils Database (Fig. 7) are more characteristic of the global fallout signature. However, along the mid-latitudes (20 to 45°S), a mixture can be

observed, with many of the points displaying shifts in the $^{240}$Pu/$^{239}$Pu atom ratios towards either the French or British fallout. Among the continents, only reference soils from the Sub-Saharan Africa showed $^{240}$Pu/$^{239}$Pu atom ratios that were exclusively attributable to the global fallout. The majority of those collected from South America and Oceania deviated from the global fallout signature with an expected significant contribution of low-yield NWT fallout in French Polynesia and Australia.

The reference soil profiles with the lowest $^{240}$Pu/$^{239}$Pu atom ratios (0.0394 ± 0.0062) characteristic of low-yield fallout were those collected from the Gambier archipelago (23° S, 135° W), which is located 425 km from the French test sites of Moruroa (22.2° S, 138.7° W) and Fangataufa (21.8° S, 138.9° W) in French Polynesia (Bouisset et al., 2021). Interestingly, comparable $^{240}$Pu/$^{239}$Pu atom ratios were also reported in reference soils collected from western Chile and Australia. Chamizo et al. (2011) reported a ratio of 0.041 ± 0.003 at a high-

altitude site in La Parva, Chile (33° S, 70° W), whereas Lal et al. (2017) reported a $^{240}$Pu/$^{239}$Pu atom ratio of 0.069 ± 0.005 in Central Australia (25.3° S, 132.0° E), which is close to the values measured in soils near the UK testing sites in the Montebello Islands (0.045 ± 0.002) and Maralinga (0.04-0.05) (Tims et al., 2013a and 2013b). All these regions therefore showed the highest contribution from either French or British fallout. For the rest of the Equatorial and Southern Hemisphere, the global fallout was considered the main source of $^{239+240}$Pu (Fig. 8).

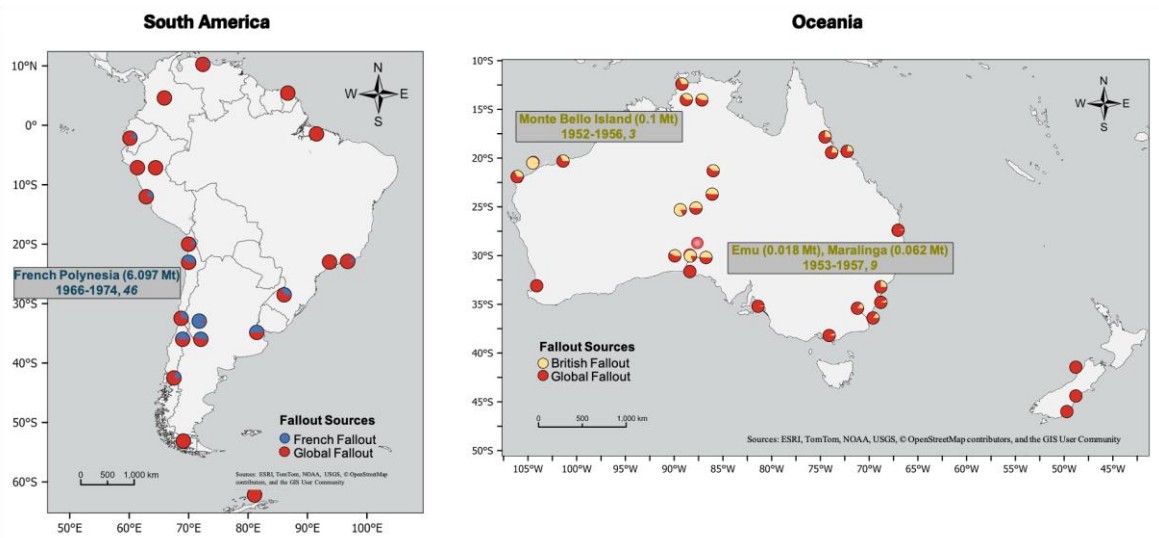

**Figure 8. Relative contribution of the fallout sources in South America and Oceania, where there is a considerable contribution of the French and British fallouts. See Fig. 4 for the exact location of testing sites. (Sources: ESRI, TomTom, NOAA, USGS, © OpenStreetMap contributors, and the GIS User Community)**

### 5.2.3 Relative contributions of fallout sources to FRN inventories

The relative contributions of the fallout sources calculated from the $^{240}$Pu/$^{239}$Pu atom ratios in South America and Oceania is shown in Fig. 8. Again, we assumed that only the contributions from the French NWT caused deviations in the ratios from the global fallout in South America and the French Polynesia, whereas the British NWT caused deviations in the rest of Oceania. The French fallout had a significant contribution between 20-40° S in South America, which is approximately the same latitudinal band where the French NWTs in Polynesia were conducted. In Australia, a significant contribution of the British fallout was observed in the central and western parts of the country, in areas located nearby and between the testing sites in Emu, Maralinga, and Monte Bello Islands at 15-35° S. On average, the French fallout contributed ~20% to the reference soil $^{239+240}$Pu inventories in South America and ~68% in French Polynesia, whereas the British fallout contributed ~27% to the reference soil $^{239+240}$Pu inventories in the rest of Oceania. The uncertainties associated with the relative contributions in each soil profile, as well as the relative contributions in areas not shown on the map, calculated using Eq. (4) can be found in Table S4. These uncertainties were calculated by propagating the uncertainties associated with the fallout source $^{240}$Pu/$^{239}$Pu atom ratios and the ratios in the soil. However, the nature of the distribution of $^{239+240}$Pu in the Southern Hemisphere could provide another source of uncertainty.

As opposed to the Northern Hemisphere, the influence of the French and British fallout in the Southern Hemisphere led to some heterogeneity in $^{240}$Pu/$^{239}$Pu atom ratios in the soil (Kelley et al., 1999; Chamizo et al., 2011; Chamizo et al., 2020). This is due to the low fallout spread over decades-long weapon development and containing significant tropospheric debris, which is less homogeneous than stratospheric fallout, integrated in the soil (Kelley et al., 1999). For example, a wide variability in $^{240}$Pu/$^{239}$Pu atom ratios were observed by Chamizo et al. (2011) in Chile within the 20-40° S range, which encompasses the latitude of the French NWT in French Polynesia. In Madagascar, a similar yet less pronounced variability was also observed in peat and marshland cores

(Chamizo et al., 2020). Interestingly, the cores with the lowest $^{239}$Pu inventories had $^{240}$Pu/$^{239}$Pu atom ratios significantly lower than the global fallout signature. However, since the studied sites are marshlands subjected to flooding regimes (Chamizo et al., 2020), factors other than atmospheric fallout likely affected the distribution of Pu in the area as well. For the same reason, the $^{210}$Pb chronology could not be established in the peat layers.

Upon comparing the $^{240}$Pu/$^{239}$Pu atom ratios analysed from 5-g aliquots of soils using thermal ionization mass spectrometry (TIMS), Kelley et al. (1999) also observed a ±16% standard deviation from the ratios previously obtained from 1-kg aliquots of the same samples by Krey et al. (1976). This is higher than the ±6% standard deviation observed in samples collected from the Northern Hemisphere (Kelley et al., 1999). This indicates that sample heterogeneity could also affect $^{240}$Pu/$^{239}$Pu atom ratios and may not fully reflect the actual contribution
from different fallout sources.

The heterogeneity associated with the contribution from different NWT sources for $^{239+240}$Pu inventories is certainly less significant than the heterogeneity associated with NPP accidents for $^{137}$Cs (Alewell et al., 2017). However, it is still critical to address the uncertainties mentioned previously, especially in areas with significant input of sources other than the global fallout when employing $^{239+240}$Pu inventories and isotopic ratios for a specific
application. This has recently been demonstrated by Dowell et al. (2024a, 2024b) in using $^{239+240}$Pu inventories to assess soil erosion in western Kenya, wherein the low variability of $^{239+240}$Pu inventories and isotopic ratios were considered as criteria for the suitability of the assessment technique. As such, the AVATAR-Soils Database provides a data compilation which might be used to assess the heterogeneities of different regions.

# 6   Predicting $^{137}$Cs and $^{239+240}$Pu inventories in Equatorial and Southern Hemisphere reference soils

The results of the Random Forest model for $^{137}$Cs and $^{239+240}$Pu are presented in Figure 9. For $^{137}$Cs, the most important covariate was precipitation during the coldest quarter, followed by longitude, annual precipitation, elevation, and latitude (Figure 9b). In the case of $^{239+240}$Pu, mean diurnal temperature range emerged as the most
important covariate, followed by precipitation during the driest quarter, temperature annual range, precipitation during the driest month, and precipitation seasonality (Figure 9d). Due to the differences in the spatial coverage of $^{137}$Cs and $^{239+240}$Pu inventory data, with $^{239+240}$Pu covering only sparse areas, different covariates emerged with different levels of importance. However, this does not imply that $^{137}$Cs and $^{239+240}$Pu are transported differently. As the $^{137}$Cs/$^{239+240}$Pu activity ratios suggest (Table 1), $^{137}$Cs and $^{239+240}$Pu appear to follow similar deposition
patterns for every fallout source.

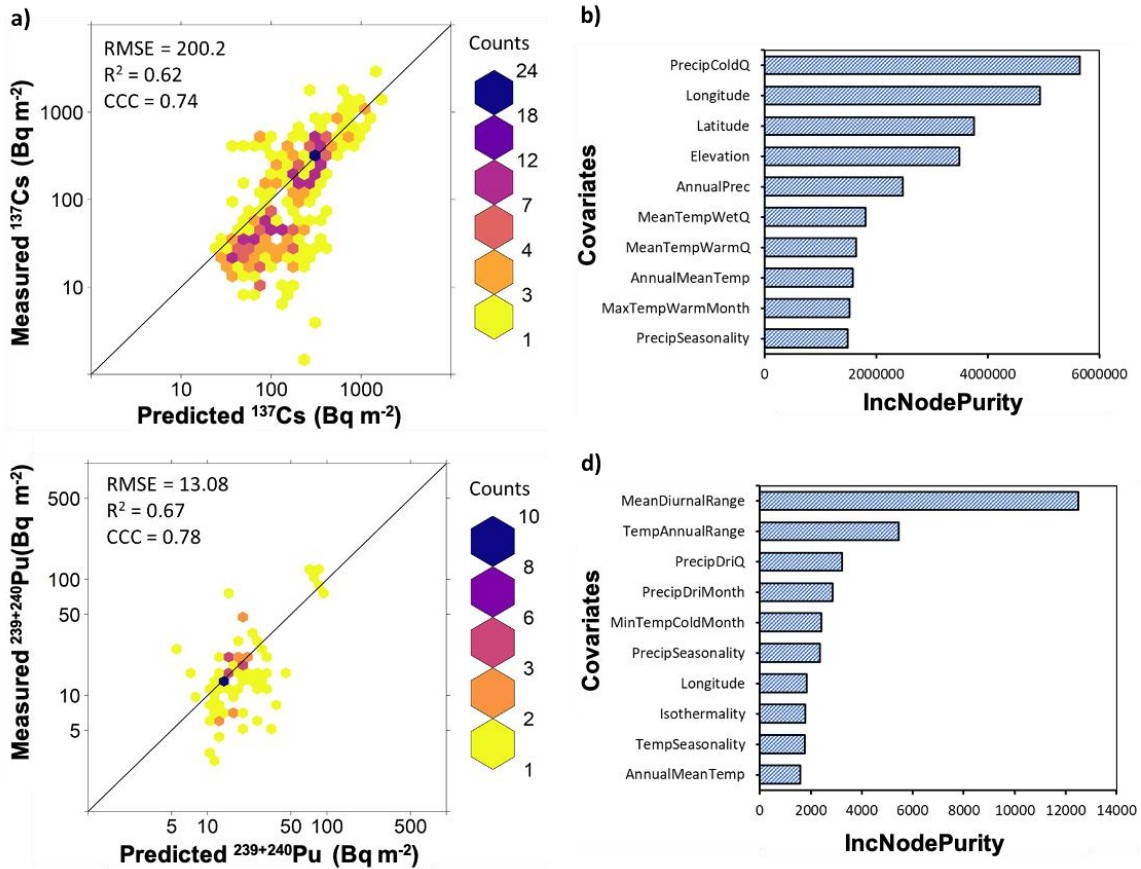

**Figure 9. Results derived from the Random Forest model. (a) and (c) Correlation between measured and cross-validation predictions of [137]Cs and [239+240]Pu, respectively. The color gradient represents the number of observations within each hexagonal bin, with the solid black line indicating the 1:1 relationship. (b) and (d) Relative importance of covariates in modeling [137]Cs and [239+240]Pu. The x-axis shows the average increase in node purity, with higher values indicating greater importance. The top 10 most significant covariates (full names listed in Table S1) are displayed in the plot.**

In addition to precipitation, spatial variables like latitude and longitude emerged as important covariates. Latitude governs the deposition of FRNs from the troposphere through latitudinally constrained atmospheric circulation cells (UNSCEAR, 2000), while longitude is related to the location of test sites in the Southern Hemisphere. Elevation and temperature-related variables also emerged as important variables, as these govern the orographic processes that influence the deposition of FRNs (Meusburger et al., 2020).

Generally, precipitation is reported to account for about 90% of FRN deposition (Wright et al., 1999) through the scavenging of particulates released from NWT (Mercer et al., 1963; Machta 1964). However, Earth's atmospheric circulation is not uniform, with wind patterns converging and diverging at different locations globally, leading to an uneven distribution of FRNs in the atmosphere. As such, the relationship between precipitation and FRN inventories is only consistent within climatologically uniform areas (Chappell et al., 2011). This is apparent from the varying correlations observed between precipitation and [137]Cs inventories in different regions like the Arctic (Wright et al., 1999; Pálsson et al., 2006), South America (Chaboche et al., 2021; Chamizo et al., 2011), and

Australia (Chappell et al., 2011). This also implies that other variables also influence the global variability of $^{137}$Cs and $^{239+240}$Pu inventories.

The optimal 'mtry' values were determined to be 8 for $^{137}$Cs and 6 for $^{239+240}$Pu. Cross-validation of the Random Forest model yielded concordance correlation coefficients (CCC) of 0.74 for $^{137}$Cs and 0.78 for $^{239+240}$Pu, with coefficients of determination ($R^2$) of 0.62 and 0.67, respectively. The root mean square errors (RMSE) were 200.2
for $^{137}$Cs and 13.08 for $^{239+240}$Pu (Figures 9a and 9c). These findings demonstrate the potential of these data for developing future baseline maps of $^{137}$Cs and $^{239+240}$Pu.

## 7 Data limitations, uncertainties, and recommendations

Compiling the data for the AVATAR-Soils database was connected to inherent data limitations that could not be
avoided due to uncertainties and obscurities in the reviewed publications. The criteria used for reference soils are often not explicitly reported, which made it challenging to verify. Thus, we report the criteria we used to define a reference soil, the vegetation at the time of sampling, and land-use history, if available. As much as possible, the coordinates of the soil profiles were taken as provided by the authors, however, the accuracy varied across different publications. While more recent publications provided highly accurate coordinates, the coordinates
provided in older publications might be connected to lower accuracy due to limitations in technology. For example, the coordinates provided in Hardy et al. (1972, 1973) were expressed only up to the tenth decimal place, resulting in some points falling outside of land surface areas when mapped in ArcGIS Pro. These points were therefore not used in the Random Forest implementation. Some older publications (e.g., Loughran et al., 1989, 1992, 1993) have also only provided maps that lack longitudinal and latitudinal axes. In these cases, the maps
were overlaied in Google Earth to extract the coordinates. A further limitation was, that some published data were only presented as figures. These figures were therefore digitized via WebPlotDigitizer v4.5 (Rohatgi, 2020), which provides only an approximation of the exact values used by the authors.

As $^{137}$Cs and $^{239+240}$Pu were measured in different laboratories using different analytical techniques (for $^{239+240}$Pu), which are connected to different measurement uncertainties, another source of uncertainty is introduced when
comparing the data compiled. In addition, the heterogeneity of samples used for analysis as pointed out by previous studies (Kelley et al., 1999; Chamizo et al. 2020) may also affect the representativeness of the values reported. Finally, $^{137}$Cs data without dates of sampling nor date of decay-correction were decay-corrected using the average time delay between sampling and publication. Users of the AVATAR-Soils database must therefore be aware of the abovementioned limitations. We also recommend that all important information related to $^{137}$Cs
and $^{239+240}$Pu be made available in future publications to increase the accuracy of the data. This includes a sufficient description of the sampling area, date of measurement or decay-correction (for $^{137}$Cs), analytical precision, uncertainties, and detection limits.

## 8 Data availability

The AVATAR-Soils Database may be accessed from the Zenodo repository:
https://doi.org/10.5281/zenodo.14008220 (Dicen et al., 2024).

## 9 Code availability

Codes used for running statistics and modelling were written in R and are available upon request from the corresponding autor.

## 10 Conclusions and outlook

Current scientific knowledge on $^{137}$Cs and $^{239+240}$Pu is better constrained in the Northern Hemisphere, and there are many uncertainties especially regarding their distribution, as well as their sources and the factors that govern their distribution, in the Southern Hemisphere. The AVATAR-Soils Database is—to the best of our knowledge— the first comprehensive compilation of $^{137}$Cs and $^{239+240}$Pu data in Equatorial and Southern Hemisphere reference soils from the literature.

The metanalysis results revealed that high $^{137}$Cs and $^{239+240}$Pu inventories were recorded near the equator and within the 20-40° S latitudinal bands, which coincide with those latitudes where many NWTs were conducted. The $^{240}$Pu/$^{239}$Pu atom ratios suggest that sources other than the global fallout (primarily from US and USSR weapons testing with a $^{240}$Pu/$^{239}$Pu atom ratio of ~ 0.18) contributed to the reference inventories in the Southern Hemisphere especially in South America and Oceania. The French fallout had a significant contribution between 20-40° S in South America, which is around the same latitude where the French NWT in Polynesia were conducted. In Australia, a significant contribution of the British fallout can be observed in the central and western parts of the country, in zones nearby and between the testing sites in Emu, Maralinga, and Monte Bello Islands. However, lake sediment data should be investigated to confirm whether or not the areas affected by the French and British fallouts, which have similar $^{240}$Pu/$^{239}$Pu atom ratios but different fallout chronologies, are indeed geographically well-delineated.

Through a modeling approach, we identified the most important set of climatic, topographic, and spatial variables that can predict the $^{137}$Cs and $^{239+240}$Pu inventories. The common predictors for both $^{137}$Cs and $^{239+240}$Pu were precipitation during the driest quarter, longitude, mean diurnal range, and other temperature-related variables. Despite the good predictability of inventories using these variables, actual inventories must still be measured in areas that lack data to further strengthen and improve the model. Through this, we will be able to develop a more comprehensive understanding of the distribution and sources of $^{137}$Cs and $^{239+240}$Pu in Equatorial and Southern Hemisphere soils and improve their application as tools in Earth Science research.

**Supplementary information**

The supplementary information is available at the onlline version of this artcle.

**Author contributions**

GD and CA conceptualized and designed the research. GD wrote the initial draft of the paper and revised the subsequent versions; SG developed the machine learning part of the study; and CA provided overall supervision.

All authors contributed by reviewing and editing the manuscript.

**Competing intesrests**

The authors have no competing interests to declare.

**Acknowledgements**

We are grateful to the researchers who generously shared their actual data used in their publications, including Adrian Chappell (Cardiff University), Stephen Tims (The Australian National University), Rajeev Lal (The Australian National University), and Gerald Raab (University of Graz). We also thank the University of Basel students Lara Brunner, Magdalena Samland, Salome Wehrli, and Linda Agnetti who dedicated time and effort to

help us extract data from the selected articles. The AVATAR Project is funded by the Swiss National Science Foundation (SNSF; Grant number 212886) and the French National Research Agency (ANR; ANR-22-CE93-0001).

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
