# Peer review of "Distribution and sources of fallout 137Cs and 239+240Pu in Equatorial and Southern Hemisphere reference soils"

_Earth System Science Data, 2024_

## Referee Comment (RC2)

Review of ESSD-2024-509, Southern Hemisphere radionuclides.

Accepting authors reasoning and accounts of FRN sources, and agreeing that current knowledge focuses more on northern than southern hemisphere, I agree with AVATAR objectives: to understand "fallout chronology and distribution of 137Cs and 239+240Pu in the Southern Hemisphere for various environmental applications". Does this database provide unique high-quality source information about these isotopes so that researchers can all 'start from the same page' in assessing southern hemisphere distributions? Should researchers trust this database to ask further questions, beyond those hinted at here? Have authors missed key information? What other links, e.g. to climate, topographic, geographic, precipitation type or quantity records, might one need, or wish for? Have authors produced a notable outcome, suitable for publication in ESSD?

"Expected". "Assumed". Readers would like to follow these authors but, absent clear uncertainty guidelines, we must remain suspicious. AVATAR database represents potentially the best state-of-the-art database of SH FRN, but authors have yet to convince this reader!

1) That we do not, no longer have, or - indeed - never had complete accurate records of tests or yields seems unfortunately not surprising. Thanks due to authors for recording and acknowledging these gaps.

2) The literature search seems appropriate but - perhaps - limited. Did authors consider other terms besides or in addition to 'soil'? 'Sediments'? 'Particulates'? 'Fall-out'? 'Precipitation'? One notes and understands extensive search on isotope terms but wonders about restricted (?) search only on 'soil'. Particularly since soil depth ('profile') and clear evidence of non-disturbance seem key? One presumes that deposition (direct or via particle run-off) into lake sediments might then offer suitable non-disturbed records? Or does 'run-off' itself represent a disqualifying erosive process? One understands desire to exclude flooding or draining, but particle run-off from soil into lake seems natural and benign. By choosing reference 'soil' exclusively, have authors missed key isotope residence sites probably experiencing relatively low disturbance? Would one expect differences in assumptions based only on soil profiles versus assumptions derived from soil profiles plus lake sediment profiles? The entire 'dry' vs 'wet' vs 'flooded' vs 'drained' categorization seems artificial but perhaps necessary? Later (line 469, line 634) authors also mention lake sediments as possible rectifiers or references for these data? Question: do lake sediment data exist in sufficient numbers, reliability and accessibility to impact initial conclusions presented here?

3) For calculating delay-induced decomposition of 137Cs during publication process, one hopes that four years represents a maximum rather than mean time. In any case, reflecting relatively short 1/2 life (fast decomposition) of 137Cs, doesn't uncertainty in publication time, e.g. 3.5 vs 4.5 years, induce subsequent uncertainties in overall 137Cs concentrations?

4) The entire latitude discussion seems - at best - mismanaged here, for several reasons. First, at these latitudes, southern hemisphere surface areas represent primarily ocean. Precipitation happens, in some areas abundantly, with substantial spatial and temporal heterogeneity, over southern hemisphere oceans. How, even using best soil profile data, can authors draw conclusions about southern hemisphere depositions or distributions by only monitoring a fraction, at some latitudes a very small fraction, of geographic surface area? Second, by their own analyses, longitude proved a more important determinant than latitude? In all cases precipitation (wet deposition) proved a determinate factor, followed by longitude; in only a few cases (for 137Cs) or in no cases (for 239+240Pu) did latitude play any statistically-important role? Third, again by authors analyses, coastal sites proved much more important to this database than 'interior' sites. Because this definition ('coastal'

vs 'interior') also varies greatly as a function of latitude, should authors have placed less (or, no) emphasis on latitude?

5) This reader remains very cautious about two-factor unmixing calculations. I credit the authors for a plausible attempt but: a) readers will eventually learn, if we have not already, that reporting of British and French (two 'local' factors) tests remain largely unknown (lines around 190); b) that documented British and French tests represent less than 30% of reported NWT contributions (line 546, 547); and c) British and French NWT occurred in distinctly different (by longitude, closeness to ITCZ, vegetation, etc.) locations. A rigorous comparison might involve NWT by two different countries from a single location? No longer possible, today, of course, but authors can predict (large) uncertainties?

6) Eventually, reader will learn that only 123 of 1526 (<10%) publications discussed outcomes that qualified for inclusion in this database. Good on authors for maintaining high standards! But, RF analyses covered these 123 publications (1122 total profiles) or the full unqualified set? RF analyses, even with 5 20% subsets run 5 times for 22 variables, in no way approaches 10000 RF runs? Or, if run for all 1100+ profile data, greatly exceeds 10k? Clearly, authors have not helped this reader understand their factor assignment process. WorldClim 2.1 only provides data on monthly averages so it will have 'smeared' precip records. Most ESSD readers avoid WorldClim data because of suspected terrestrial biases.

7) This reader offers no better alternatives but never-the-less remains very skeptical of analysis via publication records. We know publication records themselves retain substantial biases from construction and exercise. We know that Google Scholar today produces different outcomes than Google Scholar of 2021. Editors and editorial standards also change. We all know that one good 'paper' outweighs 100 weaker papers in terms of care, description, documentation, etc. We also know that publication standards for NWT or FRN data will have changed over time, from initial exploratory reports to subsequent more careful or more thoughtful analysis. The authors acknowledge such changes with their discussions of 'sequences' of "early" vs "late" papers and with their intercomparisons with prior reports. Temporal uncertainty in publication quality plays a large but largely unacknowledged role?

8) Line around 320: As n gets very low (43, 7), precision to tenths of a percent (35.0, 5.7) seems more and more fanciful?

9) Line 364, Figure 4: Map proves dominance (rightly or wrongly) of northern hemisphere sites in any and all analyses. Do authors really want to show this? Have these authors done similar quality assurance, as described here, for all northern hemisphere data?

10) Line 379, Figure 5: Figure shows very large (impracticably large) uncertainties for most southern hemisphere data, particularly when 'sorted' by supposed latitudinal bands. Uncertainties as documented in this figure negate much of the discussion?

11) Line 384, Figure 6: Figure documents difference (or, absence of statistical differences) between AVATAR and UNSCEAR data or Hardy 1973 data, but - as for prior figure - latitudinal uncertainties prove disqualifying? This reader might agree with NH vs equatorial vs SH assessments but has not seen anything so far to convince of statistically-valid latitudinal differences within SH? All evidence seems to point in opposite direction: lack of any statistically-valid latitudinal differences within SH?

12) Readers would like to trust author descriptions of NWT sources. Isotopic discrimination seems an ideal tool in this regard. But, unfortunately, too many variables cloud the authors' conclusions: location, longitude, elevation, precipitation, temporal evolution of precip and, indeed, of NWT sources, etc. Authors could greatly assist this reader by starting from, and keeping in their minds and in minds of readers, substantial uncertainties! Too easy to focus on details of isotopic analysis while forgetting larger uncertainty factors? For this reader, early declaration of a summary uncertainty, e.g. $\pm$ 90%, $\pm$50%, whatever, honored by authors throughout manuscript, would represent a substantial improvement and assistance.

13) Line 580, Figure 9: Figure attempts to show prediction skill for areas not covered by measured soil profiles. Good effort. Good luck. This reader might assume some skill for

137Cs, e.g. based on data plot plus particular but un-referenced (panel b) 'purity' factors, but finds no reason from this figure to base any predictions on 239_240Pu (panels c, d) data.

14) Uncertainty discussion refers entirely to external (weak reporting) factors. Authors apparently assume their work, or their assembly work, introduced no additional uncertainty factors. Certainly not true, but perhaps uncertainty factors introduced here remain small compacted to 'external' factors. Unfortunately, readers gain to information to buttress such conclusions?

15) Database easy to find, download, read, etc. Compliments to authors!

---

## Author Comment (AC1)

**RESPONSE TO REFEREE #1**

This paper describes a compilation of published 137Cs and 239,240Pu results in soils from the Southern Hemisphere in a database prepared under the AVATAR project. The aim of the database is to put together the scarce information on those radionuclides that is available for the southern hemisphere and use that data to understand the sources of anthropogenic radioactivity to that region using the isotopic composition of Pu. Besides, by applying a machine learning algorithm, the environmental factors that might influence the distribution of those radionuclides are identified.

Having such compilation and interpretation studies is necessary to understand reported results by different authors for a certain region and make the most of that data. Thus, the paper presents an interesting and useful study for the scientific community. The paper is correctly written and organized, and figures are representative and of high quality. However, key aspects are not properly addressed and it needs a major revision before being finally published.

We thank the reviewer for appreciating the work that we did in synthesizing the available data on 137Cs and 239+240 in Equatorial and Southern Hemisphere reference soils. We also appreciate the insightful comments, thorough review, and suggestions made on the manuscript, which we have addressed in the revised version. Responses to every comment can be found below, and changes to the manuscript have been highlighted in the manuscript file.

**General Comments**

1. Throughout the paper, the health risks of the artificial radionuclides are commented in different parts. I think such comments should not appear in this paper, which is focused on a compilation of reported data and assessment of the values. Besides, what is said about health risks of anthropogenic radioactivity is not rigorous and not realistc at all, and create a sense of drama. I refer here to the UNSCEAR 2000 report regarding the dose assessment for anthropogenic radioactivity.

   We agree that the health risks associated with artificial radionuclide contamination are not the focus of the paper. However, based on our previous experience, at conferences in particular, we are often asked about the health implications of fallout radionuclides. Therefore, to avoid misconceptions regarding the potential danger of FRN contamination, it is important to address that we are aware of this risk in the introduction. We do believe that it is appropriate to acknowledge these risks before discussing their applications. The toxicity of Pu is also highlighted by others such as Krey et al. (1976) in IAEA-SM -199/39. We also refer to the recent publications on this in the first paragraph. But we have removed the comments in other parts of the manuscript such as in the abstract and the rest of the introduction.

2. When discussing the Pu ratios to study contamination sources, the heterogenous nature of the contamination in the Southern Hemisphere should be considered. There are two papers reporting Pu ratios and inventories in soils from all over the northern and Southern Hemispheres: Hardy et al., 1972 and Kelley et al., 1999. Both papers use the same set of soil samples from all over the world, but Hardy et al., analysed aliquots of 1 kg, and Kelley of 5 g by novel techniques. Thus, different effects are observed in both studies. Indeed, Kelley et al., observed heterogenous

results (variable 240Pu/239Pu ratios) in samples from the Southern Hemisphere, far from testing sites, when analysing different aliquots from the same sample. Such effect was not observed by Hardy, since he processed 1 kg aliquots. The same heterogenous behaviour has been observed in Chamizo et al., NIMB, 2011, (doi:10.1016/j.nimb.2011.04.021) when analysing soil samples from Chile. And a similar effect was observed in peat bog cores from Madagascar in (Chamizo et al., 2020, doi.org/10.1016/j.scitotenv.2020.139993). So the fact that low ratios are not observed in a specific sample does not mean that the actual ratio is representative. It seems that in the Southern Hemisphere the Pu contamination is quite heterogenous because of the influence of low-yield French and British tests. This should be considered when discussing the sources of radionuclides and the reported results.

Thank you for pointing this out. It is indeed important to mention the heterogeneities mentioned in the literature. Thus, we have added an extensive discussion on this topic in Sec. 5.2 (Fallout Sources, Lines 574-598), citing the differences between the measurements done by Kelley et al. (1999) and Krey et al. (1976) of the samples collected by Hardy et al. (1972). We also totally agree that with many soil parameters, not only FRN, it is often important to analyse larger volumes or different sub-samples of soil than would be required from the measurement requirements. However, this is unfortunately not often possible because of the lack of sample material or the lack of time.

We have also previously indicated in Sec 7 (Data limitations, uncertainties, and recommendations) that, owing to the differences in analytical techniques (which includes differences from sampling and sample preparation up to analytical detection), the comparability of the results may be affected, which is the case with Kelley et al., 1999. However, to make this even more explicit, we have additionally indicated that sample heterogeneity could also be another factor affecting the differences in the results.

**Specific Comments**
Introduction

1. Lines 52-60: The focus of the paper is not properly presented. The introduction should start in Line 58, where the use of anthropogenic radionuclides (only the ones with a long half-life like 239Pu and 240Pu, which should be also clarified) as golden spikes of the Anthropocene is stated. The health risks of artificial radioactivity should not be mentioned in the paper. I suggest removing the reference to such health risks from the introduction and the abstract, as stated before.

   Thank you for the suggestion. We have added that only FRNs with long half-lives may provide the golden spikes (Line 57). We have also removed the reference to health risks from the abstract and other parts of the paper. However, we have kept it a part of the introduction (Lines 52-56) as explained above.

2. Line 68: 239+240Pu appears out of the blue… Please, name the Pu isotopes independently, 239Pu and 240Pu, and, at some point, state that they have been traditionally measured by alpha-spectrometry and that´s why the joined 239+240Pu activity is reported (since alpha spectrometry is not able to separate the alpha emission of both isotopes). This applied to the whole paper. Besides, when naming for the first time the different radionuclides, indicate the half-lives of them.

   Thank you. We have revised this part accordingly in Lines 66-68 and 75-78.

3. Line 74: What beta particles? Beta minus?

   Apologies for the confusion. Yes, it is beta minus with the production of metastable $^{137}$Ba. We have revised and indicated this in Line 72.

4. Line 83: The dispersion of radionuclides in the atmosphere depends on their physic-chemical properties. Caesium and Iodine are volatile and can be transported long distances. Transuranic are mainly attached to particles, and are dispersed more locally and regionally. This should be clearly explained.

   Indeed, the physico-chemical characteristics of radionuclides, especially their volatility, could affect their dispersion. This is a major factor in the case of NPP accidents such as those in Chernobyl and Fukushima wherein core meltdown temperatures were high enough (~2000 °C) to have volatilized caesium but remained below the volatilization temperature for refractory materials such as plutonium (Kirchner et al., 2011, 10.1016/j.jenvrad.2011.12.016; Steinhauser et al., 2014, 10.1016/j.scitotenv.2013.10.029). However, in the case of NWT, temperatures generated reach up to over a million degrees Celsius (Brode et al., 1964). This would then volatilize both Cs and Pu, causing them to be dispersed together, except for larger debris such as the sub-millimeter hot particles deposited near the testing sites (e.g., Cook et al., 2021, 10.1038/s41598-021-89757-5). This is also the reason why the global fallout Cs/Pu activity ratios are nearly constant through time and space as confirmed by a number of publications in Table S3, which includes the four-year fallout monitoring using air filters in ISPRA, Italy (De Bortolli et al., 1968, 10.1016/B978-1-4832-8312-8.50067-9).

   To clarify this, we have included the above information in Lines 82-85.

5. Line 93: GF peaked beginning of the 1960s.

   Thank you for the comment. However, for better readability, we intend to introduce the different sources much later in the paper, so we refer broadly to the peak of testing in the 1960s. This would also then be the basis for the calculation of how much 137Cs is left after decay.

6. Line 100: 239Pu and 240Pu have been extensively studied. They are not "new" emerging tracers anymore.

   Thank you for pointing this out. We have revised this part accordingly. Please see Lines 25-26 and 103.

7. Line 104: Note that the actual dispersion of actinides from the Chernobyl accident has not been clarified so far. There are papers reporting the presence of anthropogenic U and Pu from Chernobyl in the Baltic Sea. See, for instance, Lin et al., 2021. https://doi.org/10.1021/acs.est.1c02136

   Thank you for this information. We have revised this part of the manuscript to reflect this. Please see Lines 108-110.

Origins of fallout 137Cs and 239+240Pu in Southern Hemisphere soils

8. Line 143: Comment on the local/regional impact of the Hiroshima and Nagasaki bombs.

Comments on the Hiroshima and Nagasaki bombs have been added in the revised manuscript. Please see Lines 148-151.

9. Lines 144-145: It is mentioned that thermonuclear weapons introduced most of the FRN in the stratosphere. This should be further explained and illustrated with data from the UNSCEAR 2000 report.

Thank you for the suggestion. We have added the explanation and data in the revised manuscript. Please see Lines 159-164.

10. Line 152: Differentiate between "pure fission" atomic weapons and "thermonuclear" weapons. Is it fussion (with double s)?

We have revised this and differentiated the two types of nuclear weapons. Please see Lines 153-157.

11. Line 179: 239+240Pu differs…

We have revised this part accordingly. Please see Line 194.

12. Line 180: Sr90, Cs137 and 239,240Pu have also different geochemical properties.

This is true. However, as indicated in our response above, geochemical properties play a less significant role in the deposition of FRNs from NWT.

13. Line 192: Comment also on the fact that a significant fraction of the Southern Hemisphere is covered with water.

We have added this information in the revised manuscript.

AVATAR-Soils Database Overview

14. Line 306: Be careful when stating that Hardy et al., is the only paper reported overall Pu results. A comment on the subsequent study by Kelley et al., should be given also.

We have revised this part accordingly citing Kelley et al. in Lines 322-324.

15. Line 355: Say "the 240Pu/239Pu atom ratio". This applies to the whole paper.

Thank you for the suggestion. We have revised this accordingly throughout the manuscript.

Distribution and sources of fallout 137Cs and 239+240Pu inventories

16. Figure 4: I think quoting just the fission yield of the nuclear tests is not correct. I think the total yield of the detonations should be represented instead, which is the most reliable number we have.

Indeed, fission yields from UNSCEAR 2000 were estimated from the total yields. We have revised the figure to represent the total yields instead.

17. Line 405: Be careful when stating that only 239+240Pu results were reported by Hardy et al., 1973. The work done in Kelly et al., 1999, should be also mentioned and put in context.

We have revised this part accordingly. Please see Lines 424-425.

18. Table 1: How are the ratios for the pre-moratorium period estimated? Note that the Arctic and the Antarctic are very pristine areas that might not be comparable with soils from the general environment. You could also cite the values reported in (López-Lora et al., 2023, https://doi.org/10.1021/acs.est.2c07437) for a sediment core from the Baltic Sea where the two NWT periods are identified and studied independently.

The ratios for the pre-moratorium were determined from ice cores (Koide et al., 1979, 1982, 1985). Although these are very different from those found in soils and sediments, they are actually good sources of unaltered signatures as they only originate from atmospheric deposition. The only concern however is that they were recorded in the polar regions and receive only minimal fallout. But in the absence of available data elsewhere, these pre-moratorium ratios from ice cores are the ones we have to rely on.

Thank you for providing this resource. However, the ratios reported for the pre-moratorium and global fallout sources are almost the same as those reported by López-Lora et al., 2023 (as illustrated in Table 2 in the referred paper). This is quite different from our initial unpublished results (Guillevic et al., *in prep*) in a number of lake sediment cores from different parts of the Southern Hemisphere. We hope to clarify this in our upcoming publication of freshwater lake sediment data.

19. Table 1: Please, check the 240Pu/239Pu ratios for the Franch and British tests. The given values are very low, and might not be truly representative. For the French tests, a value of 0.05 was reported in (Chamizo et al., 2015, http://dx.doi.org/10.1016/j.nimb.2015.05.008). For the British tests, similar values were reported in (Johansen et al., 2014).

The values we reported for the French and British fallouts are lower than those reported by Chamizo et al. (2015) and Johansen et al. (2014) because lower ratios were also reported in the compiled studies in Table 1 and Table S3. In addition, this $^{240}$Pu/$^{239}$Pu signature of 0.035 ± 0.015 for the French fallout is representative of barge tests and safety trials, which resulted in local fallout at the Moruroa and Fangataufa NWT sites. We are using this signature as a reference because these are the only ones published to date. However, it is not necessarily representative of the entire French nuclear testing program, as no known signatures are available for the 37 other balloon-based nuclear tests, including the thermonuclear ones. Please see Lines 491-494.

The material IAEA-384 that was analyzed by Chamizo et al. (2015) was also been analyzed intensively by Chappini et al. (1999) via different techniques and has already been included in our compilation. For the British fallout, the compilation also already included Johansen et al. (2014). Thus, we would like to retain the values reported in Table 1 and Table S3.

20. Table 1: When reporting values for the fallout source "Global Fallout", the period 1952-1963 should be named, since soils keep the integrated signal for the whole period, and not for 1958-1963.

We totally agree that soils accumulate the integrated signal over time. However, in Table 1, we do not report the soil signals, but rather the published source signals with the recorded fallout dates of these sources (as determined from samples that

received fallout only or primarily from these sources). These signals might then be used for, e.g., unmixing or tracing.

21. Line 505: Comment on the problem of the heterogenous nature of the Pu contamination in the southern hemisphere. See the work in (Chamizo et al., 2020, https://doi.org/10.1016/j.scitotenv.2020.139993) focused on peatbogs from Madagascar. This is an evidence that deviations in the Pu ratios have been also found in Africa.

Thank you for the suggestion. Since this heterogeneity problem ultimately affects how we interpret the relative contributions of different fallout sources, we have added an extensive discussion on this topic in Sec. 5.2.3. Please see Lines 574-598.

**RESPONSE TO REFEREE #2**

Accepting authors reasoning and accounts of FRN sources, and agreeing that current knowledge focuses more on northern than southern hemisphere, I agree with AVATAR objectives: to understand "fallout chronology and distribution of 137Cs and 239+240Pu in the Southern Hemisphere for various environmental applications". Does this database provide unique high-quality source information about these isotopes so that researchers can all 'start from the same page' in assessing southern hemisphere distributions? Should researchers trust this database to ask further questions, beyond those hinted at here? Have authors missed key information? What other links, e.g. to climate, topographic, geographic, precipitation type or quantity records, might one need, or wish for? Have authors produced a notable outcome, suitable for publication in ESSD?

"Expected". "Assumed". Readers would like to follow these authors but, absent clear uncertainty guidelines, we must remain suspicious. AVATAR database represents potentially the best state-of-the-art database of SH FRN, but authors have yet to convince this reader!

We appreciate the time and effort that the reviewer dedicated to reviewing this manuscript. We hope that our answers to the reviewer's questions and concerns below will increase the clarity of our research. As this is a meta-analysis, the data base can only be as good as the original published data are. We tried our best to compile all the necessary meta data and discuss the associated uncertainties and limitations.

**Specific Comments**
1. That we do not, no longer have, or - indeed - never had complete accurate records of tests or yields seems unfortunately not surprising. Thanks due to authors for recording and acknowledging these gaps.

   Thank you for appreciating our work in highlighting the gaps in this research topic.

2. The literature search seems appropriate but - perhaps - limited. Did authors consider other terms besides or in addition to 'soil'? 'Sediments'? 'Particulates'? 'Fall-out'? 'Precipitation'? One notes and understands extensive search on isotope terms but wonders about restricted (?) search only on 'soil'. Particularly since soil depth ('profile') and clear evidence of non-disturbance seem key? One presumes that deposition (direct or via particle run-off) into lake sediments might then offer suitable non-disturbed records? Or does 'run-off' itself represent a disqualifying erosive process? One understands desire to exclude flooding or draining, but particle run-off

from soil into lake seems natural and benign. By choosing reference 'soil' exclusively, have authors missed key isotope residence sites probably experiencing relatively low disturbance? Would one expect differences in assumptions based only on soil profiles versus assumptions derived from soil profiles plus lake sediment profiles? The entire 'dry' vs 'wet' vs 'flooded' vs 'drained' categorization seems artificial but perhaps necessary? Later (line 469, line 634) authors also mention lake sediments as possible rectifiers or references for these data? Question: do lake sediment data exist in sufficient numbers, reliability and accessibility to impact initial conclusions presented here?

Thank you for your comments. The main aim of the database is to compile information on 137Cs and 239+240Pu in reference soils that have not been subjected to any kind of disturbance (e.g., major erosion, construction, digging etc.). The reason why we focus on reference soils is that we want to have information on the original fallout, not only the sources indicated by the isotopic ratios but also the total atmospheric deposition, indicated by the FRN inventories and which are critical for applications such as soil erosion assessment. We did this by thoroughly scrutinizing all publications on 137Cs and 239+240Pu contents and the isotopic ratios of soils in Equatorial and Southern Hemisphere countries.

As opposed to reference soils, lake sediments are often subject to both sedimentation processes from erosion and direct atmospheric deposition. However, lake sediments provide a great tool to reconstruct FRN deposition with time (and their variations) especially during peak deposition. Lake cores are the best archives to reconstruct the fallout chronology, whereas reference soils are only suitable for determining the integrated fallout on land. Lake cores may be used for determining the integrated fallout but only those that received minimal sedimentation from the surrounding catchment.

Complementary to the AVATAR-Soils Database, we are also preparing a compilation of lake sediment data both from the literature and our new analyses in the AVATAR Project which will be available in the upcoming publications (e.g., Guillevic et al., *in prep*) under the AVATAR Project. At present, lake sediment data are scarce, and the analysis of multilayer sediment cores is time-consuming, this is an ongoing work in the AVATAR project, but here it goes beyond the scope of the current soil compilation.

3. For calculating delay-induced decomposition of 137Cs during publication process, one hopes that four years represents a maximum rather than mean time. In any case, reflecting relatively short 1/2 life (fast decomposition) of 137Cs, doesn't uncertainty in publication time, e.g. 3.5 vs 4.5 years, induce subsequent uncertainties in overall 137Cs concentrations?

We understand the reviewer's concern and thank you for pointing this out. However, we believe that taking the mean time between sampling and publication is the most logical way to decay-correct the values. This was also done in other studies such as Chaboche et al. (2021) and Jagercikova et al. (2015).

The difference in activities/inventories due to the estimated radioactive decay of 1 year (difference between 3.5 and 4.5) is actually still lower compared to the uncertainties associated with gamma measurements of up to 10%. With this, we believe that the uncertainty introduced by estimating the time of decay-correction would not significantly impact the general patterns observed from the database. But it

would be wrong to report the values in publications with undated 137Cs inventories as they are, without an estimated decay-correction.

4. The entire latitude discussion seems - at best - mismanaged here, for several reasons. First, at these latitudes, southern hemisphere surface areas represent primarily ocean. Precipitation happens, in some areas abundantly, with substantial spatial and temporal heterogeneity, over southern hemisphere oceans. How, even using best soil profile data, can authors draw conclusions about southern hemisphere depositions or distributions by only monitoring a fraction, at some latitudes a very small fraction, of geographic surface area? Second, by their own analyses, longitude proved a more important determinant than latitude? In all cases precipitation (wet deposition) proved a determinate factor, followed by longitude; in only a few cases (for 137Cs) or in no cases (for 239+240Pu) did latitude play any statistically-important role? Third, again by authors analyses, coastal sites proved much more important to this database than 'interior' sites. Because this definition ('coastal' vs 'interior') also varies greatly as a function of latitude, should authors have placed less (or, no) emphasis on latitude?

The reviewer is correct in pointing out that the Southern Hemisphere is primarily covered with water (see our reference to this in lines 200-202). We thus limit the scope of the research to land areas and make no assumptions on FRN deposition over the oceans.

Regarding the different co-variates tested by machine-learning, this only serves as a case-study demonstrating how the database may be applied for other studies. In this case, as we also approached the distribution of FRN inventories, we used the database to determine the predictability of FNR inventories. This has implications for creating a baseline map of the Southern Hemisphere land areas, which is also a target of the AVATAR Project and something that we are already working on. In a sense, this is a preliminary work (but an important one) that would help decide which co-variates could be used for mapping.

Regarding the third comment, we do not consider "coastal sites" (along the edges of continents) as more important than interior sites. Apologies for the confusion. The fact that there are more data on coastal sites reflect only researchers' activities, previous sampling activities, and the density of human population centres in these areas and that we identify only the gaps that are certainly more prominent in the "interior area". At present, we are actually collecting information from these areas through collaborations with researchers from all over the world who send us samples or even go out sampling to obtain new material.

5. This reader remains very cautious about two-factor unmixing calculations. I credit the authors for a plausible attempt but: a) readers will eventually learn, if we have not already, that reporting of British and French (two 'local' factors) tests remain largely unknown (lines around 190); b) that documented British and French tests represent less than 30% of reported NWT contributions (line 546, 547); and c) British and French NWT occurred in distinctly different (by longitude, closeness to ITCZ, vegetation, etc.) locations. A rigorous comparison might involve NWT by two different countries from a single location? No longer possible, today, of course, but authors can predict (large) uncertainties?

We agree with the reviewer regarding the need for a rigorous comparison on the range of fallout for the British and French tests. This is also the reason why we are analysing freshwater lake sediments all over the Southern Hemisphere under the

AVATAR Project, especially in areas where the two contributions are present. The two fallout sources have different fallout timings, so stratified lake sediment cores can be used to distinguish between the two sources because of their different fallout times and thus can be recorded in different layers. Although, as the literature currently suggests, they seem to be geographically delineated. However, we have limited data which warrants further investigation.

6. Eventually, reader will learn that only 123 of 1526 (<10%) publications discussed outcomes that qualified for inclusion in this database. Good on authors for maintaining high standards! But, RF analyses covered these 123 publications (1122 total profiles) or the full unqualified set? RF analyses, even with 5 20% subsets run 5 times for 22 variables, in no way approaches 10000 RF runs? Or, if run for all 1100+ profile data, greatly exceeds 10k? Clearly, authors have not helped this reader understand their factor assignment process. WorldClim 2.1 only provides data on monthly averages so it will have 'smeared' precip records. Most ESSD readers avoid WorldClim data because of suspected terrestrial biases.

Thank you for your feedback. For Random Forest, we only used data from the final set of 123 publications compiled in the AVATAR-Soils Database.

Regarding the RF methodology, we employed five rounds of 5-fold cross-validation, , which is a commonly used approach in machine learning to improve model reliability and reduce overfitting. This resulted in 25 RF models per variable, as each fold was evaluated on different subsets of the data. While this process does not reach the scale of 10,000 RF runs, it provides a robust framework for evaluating the importance of 22 covariates across the dataset.

To assign covariate importance, we relied on the mean decrease in node purity, a standard metric in RF analysis. This metric quantifies the contribution of each covariate to reducing the model's prediction error. We selected this method because it is well-suited for identifying key predictors in ecological and environmental datasets, as also highlighted in the methods section.

We hope that this clarifies our approach to RF analysis and covariate assignment.

Regarding the use of the WorldClim dataset, we actually think that it is an invaluable resource for climate data and has helped model FRN distributions in South America (Chaboche et al., 2021) and Europe (Meusburger et al., 2020). But as pointed out in our response above, this is only a case-study relying on this meta-analysis and for this specific review meant to demonstrate the usability of the compiled database. For the ongoing actual mapping (next paper), we have gathered extensive collection of climate data from different sources as inputs to our model.

7. This reader offers no better alternatives but never-the-less remains very skeptical of analysis via publication records. We know publication records themselves retain substantial biases from construction and exercise. We know that Google Scholar today produces different outcomes than Google Scholar of 2021. Editors and editorial standards also change. We all know that one good 'paper' outweighs 100 weaker papers in terms of care, description, documentation, etc. We also know that publication standards for NWT or FRN data will have changed over time, from initial exploratory reports to subsequent more careful or more thoughtful analysis. The authors acknowledge such changes with their discussions of 'sequences' of "early" vs "late" papers and with their intercomparisons with prior reports. Temporal uncertainty in publication quality plays a large but largely unacknowledged role?

Thank you for your comments. We are aware that quality indeed varies among publications. This is also the reason why we confined our selection to Thomson Reuters Web of Science indexed journals. In addition, we set the criteria for selection and reported uncertainties in the database. We also acknowledged the uncertainties associated with the development of technology over time in Sec. 7.

8. Line around 320: As n gets very low (43, 7), precision to tenths of a percent (35.0, 5.7) seems more and more fanciful?

   We agree with the reviewer that it is not necessary to express the percentage up to the tenths for just over a hundred sample sizes and have rounded off the values. Please see Lines 335-337.

9. Line 364, Figure 4: Map proves dominance (rightly or wrongly) of northern hemisphere sites in any and all analyses. Do authors really want to show this? Have these authors done similar quality assurance, as described here, for all northern hemisphere data?

   We believe that these sites are important because they are included in the database and represent the boundaries of data in the Equatorial region covered by the current study.

10. Line 379, Figure 5: Figure shows very large (impractically large) uncertainties for most southern hemisphere data, particularly when 'sorted' by supposed latitudinal bands. Uncertainties as documented in this figure negate much of the discussion?

    The uncertainties are also a reflection of the fallout heterogeneity in the Southern Hemisphere due to the local French and British NWT.

11. Line 384, Figure 6: Figure documents difference (or, absence of statistical differences) between AVATAR and UNSCEAR data or Hardy 1973 data, but - as for prior figure - latitudinal uncertainties prove disqualifying? This reader might agree with NH vs equatorial vs SH assessments but has not seen anything so far to convince of statistically-valid latitudinal differences within SH? All evidence seems to point in opposite direction: lack of any statistically-valid latitudinal differences within SH?

    We understand the concern of the reviewer. However, for UNSCEAR data, no uncertainty has been provided in UNSCEAR (2000) for the means. To make a logical and statistically valid comparison, we bootstrapped the data from the AVATAR-Soils Database to generate a mean with a 95% confidence interval per latitudinal band. This means that if the means provided by UNSCEAR are representative of the data in soil, they would fall within the confidence interval. The same applies to Hardy et al. (1973).

12. Readers would like to trust author descriptions of NWT sources. Isotopic discrimination seems an ideal tool in this regard. But, unfortunately, too many variables cloud the authors' conclusions: location, longitude, elevation, precipitation, temporal evolution of precip and, indeed, of NWT sources, etc. Authors could greatly assist this reader by starting from, and keeping in their minds and in minds of readers, substantial uncertainties! Too easy to focus on details of isotopic analysis while forgetting larger uncertainty factors? For this reader, early declaration of a

summary uncertainty, e.g. + 90%, +50%, whatever, honored by authors throughout manuscript, would represent a substantial improvement and assistance.

We are not sure what the reviewer wants us to do here. The isotopic ratios are not influenced by any of the parameters named by the reviewer (location, longitude, elevation, precipitation, temporal evolution of precip) but merely by the fallout source. Owing to the uncertainty of the isotopic ratios of the fallout sources, we can only rely on those provided in the respective publications.

13. Line 580, Figure 9: Figure attempts to show prediction skill for areas not covered by measured soil profiles. Good effort. Good luck. This reader might assume some skill for 137Cs, e.g. based on data plot plus particular but un-referenced (panel b) 'purity' factors, but finds no reason from this figure to base any predictions on 239_240Pu (panels c, d) data.

Thank you so much for your feedback. We followed the state-of-the-art method to perform this analysis to explore whether environmental covariates can reliably predict 137Cs and 239+240Pu. As you correctly pointed out, Figures 9a and 9c present the cross-validation results for the 137Cs and 239+240Pu models, respectively.

We acknowledge that the prediction ability of 239+240Pu is less robust than that of 137Cs. This might be due to differences in spatial distribution (fewer data), measurement variability, or covariate relevance for 239+240Pu. Despite this, we believe that these initial findings are important for identifying patterns and prioritizing covariates that can be further optimized in future models.

14. Uncertainty discussion refers entirely to external (weak reporting) factors. Authors apparently assume their work, or their assembly work, introduced no additional uncertainty factors. Certainly not true, but perhaps uncertainty factors introduced here remain small compacted to 'external' factors. Unfortunately, readers gain to information to buttress such conclusions?

Unfortunately, we could not follow the reviewer's comments here and what he/she would like us to do that has not been done already. We certainly did state the uncertainties introduced such as decay-correction for undated 137Cs data.

15. Database easy to find, download, read, etc. Compliments to authors!

Thank you for your appreciation. Our goal is to have an open access database that will serve as a living document of 137Cs and 239+240Pu data in the Equatorial and Southern Hemisphere reference soils.